# Urban Socio-Semantic Segmentation with Vision-Language Reasoning

**Yu Wang**[1,2,*], **Yi Wang**[2], **Rui Dai**[2,†], **Yujie Wang**[1], **Kaikui Liu**[2],
**Xiangxiang Chu**[2], **Yansheng Li**[1,‡]

[1]Wuhan University    [2]Amap, Alibaba Group

## Abstract

As hubs of human activity, urban surfaces consist of a wealth of semantic entities. Segmenting these various entities from satellite imagery is crucial for a range of downstream applications. Current advanced segmentation models can reliably segment entities defined by physical attributes (e.g., buildings, water bodies) but still struggle with socially defined categories (e.g., schools, parks). In this work, we achieve socio-semantic segmentation by vision-language model reasoning. To facilitate this, we introduce the Urban Socio-Semantic Segmentation dataset named **SocioSeg**, a new resource comprising satellite imagery, digital maps, and pixel-level labels of social semantic entities organized in a hierarchical structure. Additionally, we propose a novel vision-language reasoning framework called **SocioReasoner** that simulates the human process of identifying and annotating social semantic entities via cross-modal recognition and multi-stage reasoning. We employ reinforcement learning to optimize this non-differentiable process and elicit the reasoning capabilities of the vision-language model. Experiments demonstrate our approach's gains over state-of-the-art models and strong zero-shot generalization. The dataset and code are open-sourced under the Apache License 2.0 at github.com/AMAP-ML/SocioReasoner.

## 1 Introduction

Urban areas, as primary hubs of human activity, are a critical subject for Earth Observation (Patino & Duque, 2013). Urban land surfaces consist of rich semantic entities, and segmenting them is crucial for downstream tasks like urban planning (Zheng et al., 2025) and environmental monitoring (Yang, 2021). These entities can be broadly grouped into two types: physical semantic entities and social semantic entities. The first encompasses entities defined by physical attributes, such as buildings, water bodies, and roads. Thanks to abundant high-resolution satellite data, current segmentation models can segment these entities precisely from visual cues in satellite imagery (Hang et al., 2022). The second comprises entities defined by social attributes, such as schools, parks, and residential districts. The identification of these entities is pivotal not only for urban analysis tasks like disease transmission (Alidadi & Sharifi, 2022), the 15-minute city (Bruno et al., 2024), but also for industrial mapping applications, such as inferring socio-semantic Areas of Interest (AOIs) from Points of Interest (POIs) to enhance navigation and recommendation (Shi et al., 2025). However, their boundaries and identities are shaped by social semantics rather than distinct visual appearances (Büttner, 2014). Since this semantic information is difficult to extract from satellite imagery alone, achieving segmentation for these socially defined entities is substantially more challenging.

Existing approaches address this challenge by incorporating auxiliary multi-modal geospatial data (e.g., Points of Interest) (Xiong et al., 2025; Zhang et al., 2017). These methods often employ separate model encoders to extract features from different modalities and train task-specific models in a fully supervised manner. However, this paradigm faces three major bottlenecks: (i) such geospatial data are often difficult to obtain due to commercial or security constraints; (ii) even when available,

---

[*]Work done during an internship at Amap.

[†]Project lead

[‡]Corresponding author.

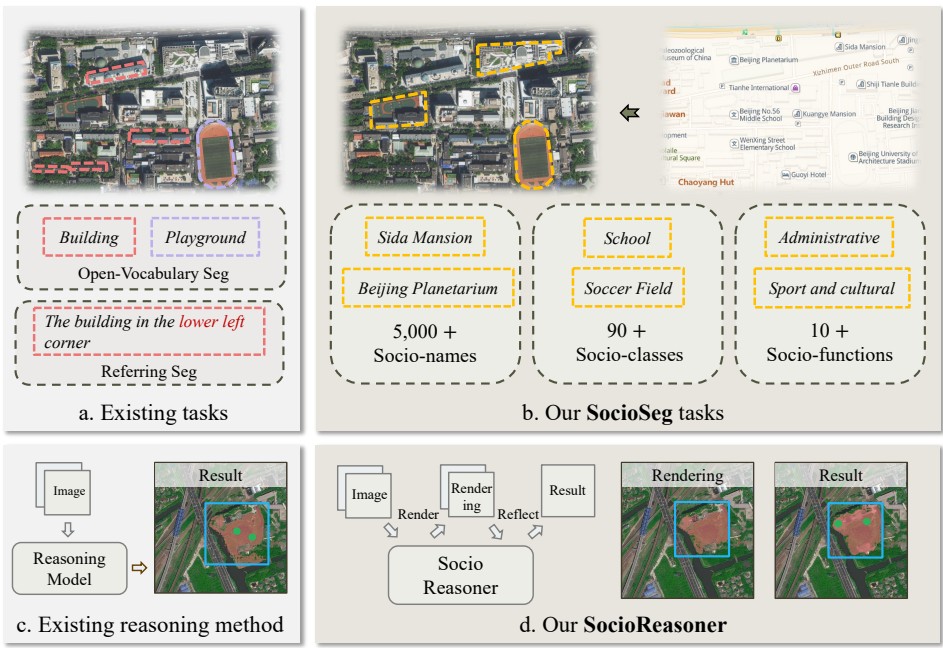

Figure 1: (a) Current works segregate physical entities. (b) Our SocioSeg identifies social entities (names, functions) via multi-modal data. (c) Existing reasoning methods employ a single-stage reasoning approach. (d) Our SocioReasoner employs a two-stage reasoning strategy with render-and-refine mechanism.

the heterogeneous formats and mismatched spatial granularities require complex preprocessing and alignment with satellite imagery; and (iii) because these methods are trained only on predefined categories, they can handle only a limited set of social semantic classes. These limitations underscore the need for a more versatile framework that can adeptly integrate diverse multi-modal geospatial data for socio-semantic segmentation.

Recent advances in Vision-Language Models (VLMs) (Achiam et al., 2023; Liu et al., 2023a; Bai et al., 2025a; Mall et al., 2024; Chu et al., 2024) offer a promising pathway toward creating such a framework. In the natural image domain, VLMs have already showcased their powerful visual understanding and reasoning capabilities on complex tasks like reasoning segmentation (Lai et al., 2024; Liu et al., 2025a; Wei et al., 2025; Chu et al., 2025), visual grounding (Bai et al., 2025b), mathematical reasoning (Zou et al., 2024), and geolocalization (Li et al., 2024a). While some work has begun applying VLMs to satellite imagery (Li et al., 2025b; Yao et al., 2025), these efforts predominantly still focus on reasoning about physical attributes. This leaves a critical gap, as social semantics, which are inherently diverse and complex, demand precisely the sophisticated reasoning processes that VLMs excel at. This natural alignment inspires us to explore the potential of VLMs for socio-semantic segmentation.

Motivated by the aforementioned challenges and opportunities, this paper defines and tackles socio-semantic segmentation by leveraging the reasoning capabilities of VLMs. To address the critical lack of a dedicated benchmark, we introduce the Urban Socio-Semantic Segmentation dataset called **SocioSeg**. SocioSeg is structured with a three-tiered hierarchy of tasks in increasing order of complexity: (i) Socio-name segmentation (e.g., "a certain university"), (ii) Socio-class segmentation (e.g., "college"), and (iii) Socio-function segmentation (e.g., "educational"). This design means the tasks place progressively higher demands on the model's reasoning abilities. Furthermore, to resolve the data-handling bottlenecks of previous methods, SocioSeg adopts a novel geospatial data representation paradigm. Instead of using raw geospatial data, which introduces the problems of access, alignment, and heterogeneity, SocioSeg unifies them into a digital map layer. This paradigm is highly effective: the need for protected raw data is eliminated, and the map layer is inherently spatially aligned with the satellite imagery.

Building on SocioSeg, we propose **SocioReasoner**, a vision-language reasoning framework that simulates the human process of identifying and annotating socio-semantic entities through cross-

modal recognition and multi-stage reasoning. More specifically, given a textual instruction with socio-semantic concepts, SocioReasoner employs a two-stage reasoning strategy with a render-and-refine mechanism: it first generates bounding box prompts from both satellite and map imagery to localize the target region. These prompts are then fed to the Segment Anything Model (SAM) (Ravi et al., 2024) to produce an initial coarse segmentation. Recognizing that segmentation from a bounding box alone can be imprecise and inconsistent with the actual human annotation process, SocioReasoner proceeds to generate point prompts on the rendered mask to refine the boundary, ultimately generating a high-fidelity segmentation result. This entire interactive process closely mirrors the workflow of a human annotator. Since this pipeline is non-differentiable, we employ a popular reinforcement learning algorithm, GRPO (Shao et al., 2024; Guo et al., 2025), to train the framework end-to-end, which also effectively elicits the VLM's latent reasoning capabilities for the social semantic segmentation task. Extensive experiments show that our approach outperforms state-of-the-art segmentation baselines and exhibits strong zero-shot generalization capabilities, highlighting the potential of combining satellite imagery with rendered map context for social semantic understanding. In summary, our contributions are:

- We introduce socio-semantic segmentation, a novel and challenging segmentation task, and release the benchmark SocioSeg, which establishes the paradigm of rendering heterogeneous geospatial data into a unified map image, transforming a complex multi-modal challenge into a visual reasoning task.

- We propose SocioReasoner, a segmentation framework that mimics human annotation via a multi-stage reasoning process. This non-differentiable workflow is optimized using reinforcement learning with a dedicated reward function, effectively eliciting the model's reasoning capabilities.

- Extensive empirical evidence demonstrates the effectiveness and generalization capabilities of our approach, highlighting its potential for real-world applications.

## 2 RELATED WORK

### 2.1 SEMANTIC SEGMENTATION

Semantic segmentation is a fundamental task in computer vision (Voulodimos et al., 2018). Early deep learning methods trained models in a fully supervised manner, enabling them to recognize only a predefined set of semantic categories (Ronneberger et al., 2015; Xie et al., 2021; Zhang et al., 2022). With the advancement of pre-trained models, tasks such as open-vocabulary segmentation (Ghiasi et al., 2022) and referring segmentation (Wang et al., 2022) have emerged, allowing models to identify unseen categories or segment objects based on textual descriptions. More recently, the task of reasoning segmentation (Lai et al., 2024) is introduced, where the input text describes the target's function or relationship rather than its visual appearance. This demands more sophisticated reasoning capabilities from the model. Notably, a significant body of current work now employs VLM-based paradigms to address reasoning segmentation tasks (Liu et al., 2025a; You & Wu, 2025; Liu et al., 2025b). These methods feed visual prompts (e.g., bounding boxes or points) derived from VLM inference into the SAM to perform segmentation, and employ reinforcement learning to elicit the model's reasoning capabilities.

Semantic segmentation from satellite imagery follows a similar developmental trajectory (Kotaridis & Lazaridou, 2021). It began with fully supervised models for extracting features like buildings (Cheng et al., 2019) and roads (Sun et al., 2019), and has since progressed to explorations in open-vocabulary (Li et al., 2025a; Zhu et al., 2025) and referring segmentation (Chen et al., 2025). Recently, some studies also begin to tackle reasoning segmentation on satellite imagery, often by using closed-source vision language model to re-frame existing segmentation categories into text that requires reasoning (Li et al., 2025b). This existing work predominantly focuses on categories defined by physical attributes (e.g., buildings, water bodies) or categories with distinct visual features. Socio-semantic categories (e.g., schools, parks), whose boundaries and identities are determined more by social constructs than by distinct visual cues, remain a significant challenge for methods that rely solely on satellite imagery. In contrast to existing work, our paper specifically targets these socio-semantic categories within urban regions.

## 2.2 MULTI-MODAL APPROACHES FOR URBAN UNDERSTANDING

The task of segmenting urban social semantic entities, which we term urban socio-semantic segmentation, is a nascent research area. While no prior work directly addresses this task, related problems exist in the field of urban science, such as land-use classification (Xiong et al., 2025) and urban functional zone (Yao et al., 2018). These studies typically fuse multimodal data, such as Points of Interest (POIs) and road networks, with satellite imagery. Their common approach involves using separate model encoders for different data modalities and then merging the extracted features for classification or segmentation (Xiong et al., 2025; Zhang et al., 2017). However, these methods, which rely on raw multi-modal data, face several critical bottlenecks. They are often hampered by challenges in data acquisition (due to commercial or security constraints), the complexity of handling heterogeneous data formats and mismatched spatial granularities, and an inability to generalize beyond a limited set of predefined categories. Crucially, this highlights the fundamental difference between our task and traditional land-use classification. While the latter typically targets a fixed, closed set of categories, our socio-semantic segmentation involves fine-grained attributes (over 90 categories) and specific entity names, where each instance acts as a unique class. Consequently, our task aligns more closely with open-vocabulary, referring, or reasoning segmentation rather than standard classification.

## 3 SOCIOSEG DATASET

Existing semantic segmentation dataset (Wang et al., 2021; Li et al., 2024b) from satellite imagery has been largely confined to extracting entities defined by physical attributes. To expand the scope to social semantics, we introduce the SocioSeg dataset, which is distinguished by two key features:

**Hierarchical Socio-Semantic Segmentation Task Design**. As illustrated in Appendix A.1.1, Figure 6, we define urban socio-semantic entities across three hierarchical levels of increasing abstraction and difficulty: Socio-names (e.g., "a certain university"), Socio-classes (e.g., "college"), and Socio-functions (e.g., "educational"). This tiered structure facilitates a progressive evaluation of a model's reasoning capabilities. Above all, SocioSeg is exceptionally rich in social semantic information, containing over 5,000 Socio-names, 90 Socio-classes, and 10 Socio-functions.

**Multi-Modal Data with Digital Map Representation**. A key innovation of the SocioSeg dataset is its unification of diverse geospatial information into a single digital map layer. This representation offers several distinct advantages. First, it overcomes data accessibility issues, as publicly available map layers replace raw multi-modal data that are often proprietary or restricted. Second, the map layer is inherently co-registered with the satellite imagery, which eliminates the need for complex data alignment. Finally, this fusion into a single visual modality provides rich socio-semantic cues that are crucial for enhancing a model's social reasoning capabilities.

We construct the inputs for SocioSeg by sourcing satellite images and digital maps from the Amap public API[1], which provides these maps in both Chinese and English versions. The digital maps render only basic geospatial information, including roads and points of interest. We then collected the ground-truth socio-semantic labels for the corresponding regions. (Further details on the annotation procedure and dataset statistics are available in Appendix A.1.1). As a result, the SocioSeg dataset comprises over 13,000 samples distributed across the three hierarchical tasks. Each sample consists of a satellite image, a digital map, and a corresponding socio-semantic mask label. We partitioned the dataset into training, validation, and test sets using a 6:1:3 ratio, ensuring that the sample counts and class distributions for each hierarchical task are consistent across all splits.

## 4 SOCIOREASONER FRAMEWORK

### 4.1 HUMAN-LIKE REASONING SEGMENTATION PROCESS

Prevailing reasoning-segmentation methods (Liu et al., 2025b; Yao et al., 2025) typically follow a single-stage pipeline: a Vision-Language Model (VLM) generates visual prompts (e.g., a bounding box), which are then fed into a frozen SAM to produce the final mask. Because the weights of

---

[1]Amap API Documentation. https://lbs.amap.com/. Accessed: 2025-05-14.

SAM are fixed, these methods lack direct control over the output quality, often resulting in coarse or inaccurate segmentation. In contrast, our SocioReasoner framework (as shown in Figure 2) employs a two-stage reasoning strategy with a render-and-refine mechanism to emulate the sequential workflow of a human annotator. This multi-stage approach enhances precision and makes the model's inference steps transparent and interpretable.

**Stage-1 (Localization): Emitting a set of 2D bounding boxes.** Let the VLM be denoted by $\mathcal{F}$. Given a satellite image $\mathbf{I}_s$, a digital map $\mathbf{I}_m$, and a textual instruction $\mathbf{t}_b$, the VLM emits a set of 2D bounding boxes $\mathcal{B} = \{\mathbf{b}_i\}_{i=1}^{N}$ to localize candidate target regions:

$$\mathcal{B} = \mathcal{F}(\mathbf{I}_s, \mathbf{I}_m, \mathbf{t}_b). \tag{1}$$

These bounding boxes are supplied to a pre-trained segmentation model, SAM ($\mathcal{S}$) (Ravi et al., 2024), to produce a preliminary coarse mask $\mathbf{M}_c$:

$$\mathbf{M}_c = \mathcal{S}(\mathbf{I}_s, \text{prompt} = \mathcal{B}). \tag{2}$$

**Stage-2 (Refinement): Emitting both a set of bounding boxes and points.** Recognizing that segmentation from bounding boxes alone can be imprecise, we provide visual feedback to the VLM by rendering both the boxes and the coarse mask onto the inputs. A rendering function $\mathcal{D}$ overlays $\mathcal{B}$ and $\mathbf{M}_c$ onto the satellite image $\mathbf{I}_s$ and the digital map $\mathbf{I}_m$, producing a pair of rendered images $(\mathbf{I}_{s,r}, \mathbf{I}_{m,r})$ for re-evaluation:

$$\mathbf{I}_{s,r} = \mathcal{D}(\mathbf{I}_s, \mathcal{B}, \mathbf{M}_c), \quad \mathbf{I}_{m,r} = \mathcal{D}(\mathbf{I}_m, \mathcal{B}, \mathbf{M}_c). \tag{3}$$

Conditioned on $(\mathbf{I}_{s,r}, \mathbf{I}_{m,r})$ and the instruction $\mathbf{t}_p$, the VLM emits a set of bounding boxes $\mathcal{B}$ together with points $\mathcal{P} = \{\mathbf{p}_j\}_{j=1}^{K}$:

$$\{\mathcal{B}, \mathcal{P}\} = \mathcal{F}(\mathbf{I}_{s,r}, \mathbf{I}_{m,r}, \mathbf{t}_p). \tag{4}$$

Finally, the comprehensive set of prompts (bounding boxes and points) is fed back into SAM to yield the final mask $\mathbf{M}_f$:

$$\mathbf{M}_f = \mathcal{S}(\mathbf{I}_s, \text{prompt} = \{\mathcal{B}, \mathcal{P}\}). \tag{5}$$

By decomposing the segmentation challenge into this sequence of localization and refinement, SocioReasoner achieves superior accuracy and provides an explicit reasoning chain. As this entire pipeline is non-differentiable, we leverage reinforcement learning to optimize the VLM's policy for generating these sequential prompts.

## 4.2 END TO END REINFORCEMENT LEARNING OPTIMIZATION

We optimize the non-differentiable, multi-stage prompting policy of SocioReasoner using reinforcement learning with Group Relative Policy Optimization (GRPO) (Guo et al., 2025). A single Vision-Language Model (VLM) policy is shared across both stages and emits structured textual outputs that encode prompts for SAM. The environment parses these outputs, executes SAM with the parsed prompts, and returns a scalar reward.

**Stage-1 (Localization) Optimization.** Given an input $\mathbf{x}_1 = (\mathbf{I}_s, \mathbf{I}_m, \mathbf{t}_b)$, the policy $\pi_\theta$ stochastically generates a completion $\mathbf{y}_1$ that encodes a set of bounding boxes. The environment parses $\mathbf{y}_1$ to obtain $\mathcal{B}$, runs SAM to produce a coarse mask $\mathbf{M}_c$, and returns a stage-1 reward $R_1(\mathbf{y}_1; \mathbf{x}_1)$ comprising: (i) a binary syntax reward to ensure valid JSON output, (ii) a localization accuracy term for the predicted boxes, and (iii) a reward for matched object count. GRPO is applied per input by drawing $G$ completions $\{\mathbf{y}_1^{(g)}\}_{g=1}^{G}$, computing rewards $\{R_1^{(g)}\}_{g=1}^{G}$, and defining a group-relative baseline $b_1(\mathbf{x}_1) = \frac{1}{G} \sum_{g=1}^{G} R_1^{(g)}$, with advantages $A_1^{(g)} = R_1^{(g)} - b_1(\mathbf{x}_1)$. The stage-1 objective is a clipped PPO-like surrogate with KL regularization against a frozen reference policy $\pi_{\text{ref}}$:

$$\begin{aligned}
\mathcal{L}_1(\theta) &= -\frac{1}{G} \sum_{g=1}^{G} \sum_{t \in \mathcal{I}(\mathbf{y}_1^{(g)})} \min\left(r_{1,t}^{(g)} A_1^{(g)}, \; \text{clip}(r_{1,t}^{(g)}, 1-\epsilon, 1+\epsilon) A_1^{(g)}\right) \\
&\quad + \beta \, \text{KL}\left(\pi_\theta(\cdot|\mathbf{x}_1) \, \| \, \pi_{\text{ref}}(\cdot|\mathbf{x}_1)\right),
\end{aligned} \tag{6}$$

where $r_{1,t}^{(g)} = \frac{\pi_\theta\left(y_{1,t}^{(g)} \mid y_{1,<t}^{(g)}, \mathbf{x}_1\right)}{\pi_{\theta_{\text{old}}}\left(y_{1,t}^{(g)} \mid y_{1,<t}^{(g)}, \mathbf{x}_1\right)}$ is the token-level importance ratio. The hyperparameters $\epsilon$ and $\beta$ control the PPO clipping and KL regularization, respectively.

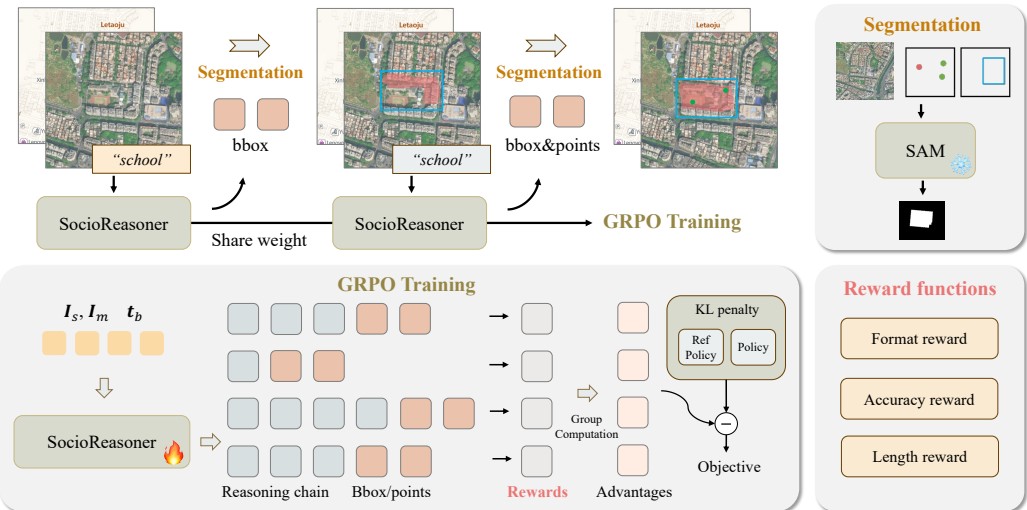

Figure 2: SocioReasoner Framework. Given a satellite image, a digital map, and a textual instruction, the VLM first generates bounding boxes to localize candidate regions. These boxes are fed into SAM to produce a coarse mask. The boxes and mask are then rendered onto the inputs for re-evaluation. The VLM emits boxes and points, which are again fed into SAM to yield the final mask.

**Stage-2 (Refinement) Optimization.** Conditioned on the rendered visual feedback and the coarse mask, the policy refines the prompts. We construct $\mathbf{x}_2 = (\mathbf{I}_{s,r}, \mathbf{I}_{m,r}, \mathbf{t}_p, \mathbf{M}_c)$ by overlaying the stage-1 boxes and coarse mask using the renderer $\mathcal{D}$. The policy $\pi_\theta$ emits $\mathbf{y}_2$ that encodes bounding boxes and points. The environment parses $\mathbf{y}_2$ to obtain $\{\tilde{\mathcal{B}}, \mathcal{P}\}$, runs SAM to produce the final mask $\mathbf{M}_f$, and returns a stage-2 reward $R_2(\mathbf{y}_2; \mathbf{x}_2)$ comprising: (i) a binary syntax reward for valid JSON, (ii) a pixel-level IoU term for $\mathbf{M}_f$, and (iii) a reward for point length. GRPO sampling, baseline/advantage computation, and the clipped surrogate with KL regularization follow the same formulation as in stage-1.

**Training Schedule.** Within a single reinforcement learning step, we execute both stages sequentially: (i) sample, evaluate, and update with $\mathcal{L}_1(\theta)$ using stage-1 rollouts; (ii) construct the stage-2 inputs from the stage-1 outputs, then sample, evaluate, and update with $\mathcal{L}_2(\theta)$. This two-stage procedure aligns optimization with the sequential localization–refinement workflow. Detailed formulations of the rewards $R_1$ and $R_2$ are provided in the Appendix A.2.2. The overall training algorithm is summarized in Algorithm 1.

## 5 EXPERIMENTS

### 5.1 BASELINES AND EVALUATION METRICS

We primarily compare against three families of methods: (i) standard semantic segmentation models, including the CNN-based UNet (Ronneberger et al., 2015) and the Transformer-based Seg-Former (Xie et al., 2021); (ii) state-of-the-art reasoning segmentation for natural images, including VisionReasoner (Liu et al., 2025b), Seg-R1 (You & Wu, 2025), and SAM-R1 (Huang et al., 2025); (iii) state-of-the-art satellite image segmentation methods, including the open-vocabulary segmentation SegEarth-OV (Li et al., 2025a), referring segmentation RSRefSeg (Chen et al., 2025), and reasoning-based approaches SegEarth-R1 (Li et al., 2025b) and RemoteReasoner (Yao et al., 2025). Because SocioSeg provides two images (satellite and digital map), we adapt all VLM-based baselines to accept dual-image inputs; for methods (RSRefSeg and SegEarth-R1) that do not support multiple images, we provide only the satellite image. *All baselines are re-trained on the SocioSeg training split to ensure fair comparison.* Additionally, given the challenging nature of the SocioSeg benchmark, we employ off-the-shelf Large Multimodal Models as reference models to provide a

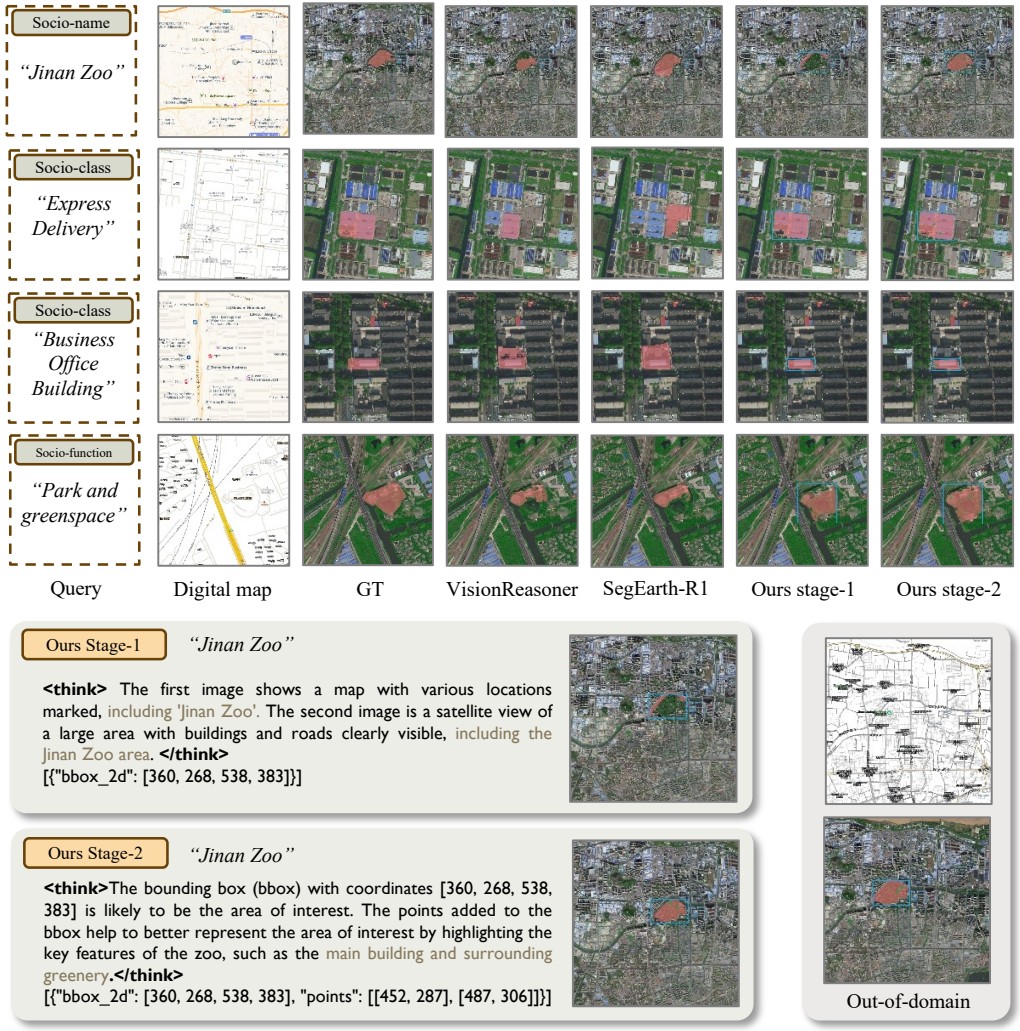

Figure 3: Visualization of the SocioReasoner results. The top panel shows a comparison between the results of SocioReasoner (with both stages visualized) and competitive baselines. The bottom-left panel illustrates the reasoning process of SocioReasoner. The bottom-right panel displays the visualization results of SocioReasoner on the out-of-domain dataset.

more comprehensive evaluation. Detailed experimental results for these models are provided in the Appendix A.4. For evaluation, we follow previous work (Lai et al., 2024) in reporting cIoU and gIoU. Additionally, we employ the F1 score to assess instance-level performance.

## 5.2 COMPARISON WITH STATE-OF-THE-ART METHODS

Comparison with state-of-the-art methods on the SocioSeg test set is presented in Figure 3 and Appendix A.6. The quantitative results are presented in Table 1, with results grouped by task for clarity. Our SocioReasoner framework consistently outperforms all baselines across all three hierarchical tasks, demonstrating its effectiveness in handling the complexities of socio-semantic segmentation. This performance gain underscores the advantage of our human-like reasoning process and the use of rendered map context in enhancing the model's understanding of social semantics. However, because SocioReasoner simulates a multi-step human reasoning process, its inference time is longer compared to other methods. We provide a detailed analysis of SocioReasoner's inference time in Appendix A.5.1. Additionally, we illustrate the accuracy for individual Socio-classes in Figure 4. Our method consistently outperforms baselines on the top-20 most frequent socio-class categories. Re-

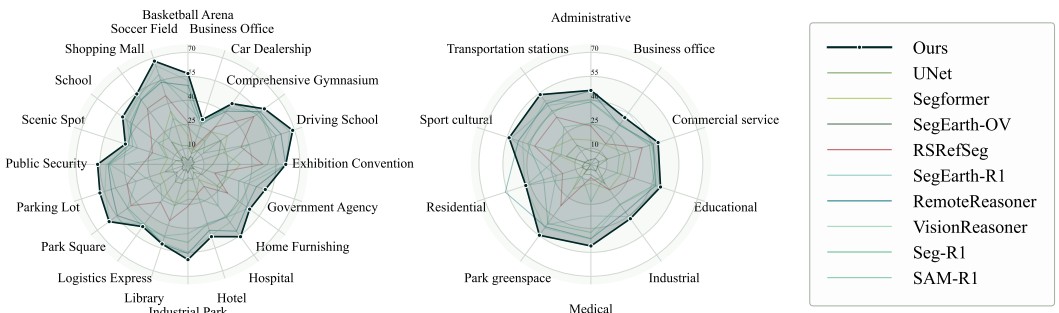

Figure 4: Per-class accuracy comparison across Socio-classes and Socio-functions. We select the top-20 most frequent Socio-classes (left) and all Socio-functions(right) for visualization.

garding the socio-function tasks, our approach achieves state-of-the-art results across all categories, with the exception of *Business Office* and *Residential*. We attribute these failures to error propagation: in certain samples, the initial bounding box localization (Stage-1) deviates significantly from the ground truth. Consequently, the point prompts generated during the refinement phase (Stage-2) tend to exacerbate rather than correct this initial deviation, as illustrated in the failure cases in Figure 12.

**Comparison with standard semantic segmentation methods.** Since standard semantic segmentation models are incapable of processing multimodal inputs, they fail to perceive the social semantic information inherent in the SocioSeg task. Consequently, under this setting, the task for standard models degenerates into a binary classification problem. As observed, UNet and SegFormer exhibit inferior performance on SocioSeg compared to the other two categories of methods, whereas our approach significantly outperforms them. Additionally, we tested the performance of Socio-class and Socio-function under a multi-class semantic segmentation setting. As shown in Table 10, because social semantic categories lack distinct visual features in satellite images, standard semantic segmentation methods perform even worse under this setting, further highlighting the challenging nature of the SocioSeg task.

**Comparison with natural image reasoning segmentation methods.** Similar to SocioReasoner, VisionReasoner, Seg-R1, and SAM-R1 all support multi-image inputs and therefore perform relatively well on SocioSeg. Notably, SAM-R1 (Huang et al., 2025) lacks constraints on the length of the output point prompts; in our reproduction, it emits a large number of point coordinates, which degrades performance. These methods freeze SAM parameters and perform single-stage inference. In contrast, our SocioReasoner framework surpasses these methods by a notable margin across all metrics. This improvement is attributable to our multi-stage reasoning process that mimics human annotation, providing reflection and refinement capabilities that lead to more accurate segmentation.

**Comparison with advanced satellite image segmentation methods.** SegEarth-OV completely freezes the CLIP encoder, limiting its recognition capabilities to the categories present in CLIP's pre-training data. Since SocioSeg features novel semantic categories related to social attributes, this method fails to function effectively on the SocioSeg task. RSRefSeg and SegEarth-R1, which are designed for segmenting physical attributes and support only a single satellite image input, show limited performance on socio-semantic tasks. However, because they are trained in a fully supervised manner without freezing the mask decoder, they achieve some performance gains. In contrast, our approach leverages multimodal reasoning, effectively integrating satellite imagery with digital map context to capture nuanced social semantics. RemoteReasoner adopts a design similar to VisionReasoner, supports multi-image inputs, and performs well on SocioSeg. Our SocioReasoner framework outperforms RemoteReasoner, highlighting the benefits of our two-stage localization and refinement process, which enables more precise segmentation through iterative reasoning.

## 5.3 ABLATION STUDIES

We ablate three core design choices of SocioReasoner: the training/inference scheme (single-stage vs. two-stage), the impact of reinforcement learning (RL), and the number of points issued in the

| Method | Socio-name | | | Socio-class | | | Socio-function | | | All dataset | | |
|---|---|---|---|---|---|---|---|---|---|---|---|---|
| | cIoU | gIoU | F1 | cIoU | gIoU | F1 | cIoU | gIoU | F1 | cIoU | gIoU | F1 |
| UNet | 10.9 | 9.4 | 8.0 | 12.6 | 11.9 | 11.2 | 11.1 | 10.6 | 10.4 | 11.7 | 10.7 | 10.0 |
| Segformer | 22.0 | 19.6 | 18.1 | 22.4 | 21.4 | 19.5 | 21.4 | 20.2 | 17.9 | 22.1 | 20.5 | 18.7 |
| VisionReasoner | 48.5 | 50.9 | 58.4 | 44.4 | 49.3 | 55.5 | 36.3 | 41.8 | 45.0 | 44.0 | 48.5 | 54.3 |
| Seg-R1 | 46.0 | 48.1 | 50.4 | 40.4 | 44.7 | 45.2 | 34.5 | 39.5 | 36.5 | 41.0 | 45.0 | 45.2 |
| SAM-R1 | 25.6 | 25.4 | 37.2 | 22.3 | 23.8 | 32.1 | 17.7 | 19.9 | 25.2 | 22.5 | 23.7 | 32.4 |
| SegEarth-OV | 3.3 | 3.3 | 0.0 | 3.8 | 3.8 | 0.0 | 4.2 | 4.2 | 0.0 | 3.7 | 3.7 | 0.0 |
| RSRefSeg | 27.1 | 25.4 | 30.9 | 30.7 | 30.6 | 35.3 | 28.7 | 28.8 | 30.8 | 29.0 | 28.3 | 32.8 |
| SegEarth-R1 | 36.9 | 42.1 | 46.9 | 38.9 | 45.1 | 50.0 | 39.5 | 45.6 | 47.4 | 38.3 | 44.1 | 48.4 |
| RemoteReasoner | 46.6 | 49.5 | 56.1 | 42.9 | 48.0 | 53.9 | 38.0 | 43.5 | 47.2 | 43.2 | 47.7 | 53.3 |
| **Ours** | **52.6** | **55.7** | **64.6** | **47.6** | **52.8** | **60.1** | **40.6** | **46.9** | **50.3** | **47.9** | **52.8** | **59.7** |

Table 1: Comparison with state-of-the-art methods on SocioSeg test set, split by task groups for readability. The best performance in each column is highlighted in **bold**. The second best is underlined. Baselines are re-trained on the SocioSeg training split to ensure fair comparison.

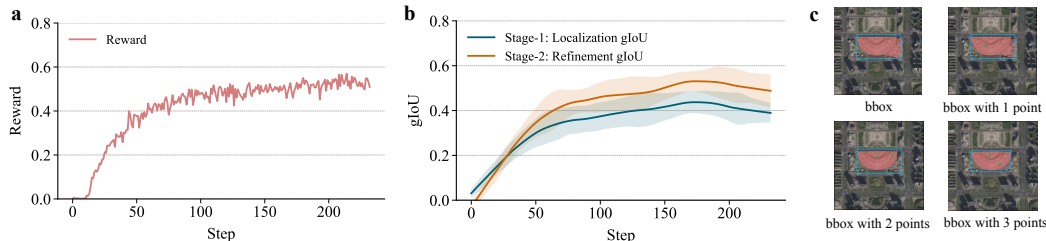

Figure 5: (a) Sum reward during training. It shows the sum of rewards across training steps in the two-stage workflow. (b) Multi-stage gIoU during training. It shows the gIoU improvement across training steps in the two-stage workflow. (c) Different number of points. It visualizes the result of SocioReasoner in the refinement stage with different numbers of points.

second stage. Results are summarized in Table 2, Table 3, and Table 4. The full results of each ablation setting are provided in Appendix A.5.1.

**Impact of the training/inference scheme.** In the "w/o reflection" configuration, the model bypasses the two-stage workflow and instead produces bounding boxes and points in a single stage, equivalent to VisionReasoner's one-step prompting. This setting performs the worst for two reasons: (i) without an iterative process, the model cannot self-correct after observing the coarse mask; and (ii) it must solve a complex planning-and-parsing problem in one shot (jointly synthesizing boxes and points in a long structured output), which increases failure rates. In the "w/o refinement" ablation, we use the model trained with the two-stage pipeline but halt the inference process after Stage-1. The output from this initial localization stage is used directly as the final result, completely bypassing the refinement stage. The complete pipeline ("Ours"), which overlays stage-1 outputs and emits both boxes and points, achieves the best results. Figure 5b shows the evolution of mask IoU across the two stages during RL training: stage-1 accuracy is initially higher because the model focuses more on localization early on; as training progresses, the model increasingly leverages points to improve the mask, leading to a steady rise in stage-2 accuracy. This finding highlights the effectiveness of our multi-stage reasoning process, where the refinement stage contributes to enhancing segmentation quality.

**Impact of the number of points in the refinement stage.** In our reward function, the parameter $\mu$ directly controls the number of point prompts generated in the refinement stage. We present the experimental results for different numbers of points in Table 3 and visualize the corresponding qualitative results in Figure 5c. We observe that a single point prompt often fails to cover the entire target, while the model struggles to learn a stable distribution for three points, with marginal performance gains compared to using two. Therefore, we select two points as the final design choice.

Table 2: Ablation of multi-stage design.

| Method | All dataset | | |
|---|---|---|---|
| | cIoU | gIoU | F1 |
| w/o reflection | 44.0 | 48.5 | 54.3 |
| w/o refinement | 46.4 | 50.8 | 57.5 |
| Ours | **47.9** | **52.8** | **59.7** |

Table 3: Ablation of point number.

| Method | All dataset | | |
|---|---|---|---|
| | cIoU | gIoU | F1 |
| 1 point refinement | 47.6 | 51.2 | 58.0 |
| 2 points refinement | 47.9 | **52.8** | **59.7** |
| 3 points refinement | **48.9** | 52.3 | 58.8 |

Table 4: Generalization of SocioReasoner, where ID and OOD refer to in-domain and out-of-domain, respectively.

| Method | ID | | | OOD (Map Style) | | | OOD (New Region) | | |
|---|---|---|---|---|---|---|---|---|---|
| | cIoU | gIoU | F1 | cIoU | gIoU | F1 | cIoU | gIoU | F1 |
| VisionReasoner (SFT) | 44.1 | 47.2 | 52.1 | 38.8 | 40.9 | 45.5 | 22.5 | 24.7 | 26.1 |
| VisionReasoner (RL) | 44.0 | 48.5 | 54.3 | _42.0_ | _44.4_ | _51.2_ | _32.8_ | _34.4_ | _35.0_ |
| Ours (SFT) | _47.1_ | _51.4_ | _57.8_ | 39.7 | 42.0 | 46.9 | 30.1 | 32.3 | 31.5 |
| Ours (RL) | **47.9** | **52.8** | **59.7** | **45.1** | **49.1** | **57.7** | **40.2** | **43.4** | **42.9** |

**Impact of the RL.** Figure 5a illustrates the reward trajectory during training. The consistent upward trend demonstrates that RL effectively optimizes SocioReasoner's human-like workflow. To further demonstrate the effectiveness of our training strategy, we compare SocioReasoner trained via RL against a Supervised Fine-Tuning (SFT) baseline. We evaluate performance not only on the in-domain (ID) Amap data but also on two challenging out-of-domain (OOD) scenarios using Google Maps tiles, as reported in Table 4. The "OOD (Map Style)" setting tests robustness to cartographic style shifts. Furthermore, we introduce a specific "OOD (New Region)" setting evaluated on a newly constructed dataset sampled from five global cities: Tokyo (Asia), New York (North America), São Paulo (South America), London (Europe), and Nairobi (Africa). This dataset, which is detailed in Appendix A.1.2, comprises 3,200 samples covering 80 categories, including 24 novel classes unseen during training. While the SFT baseline suffers significant performance degradation on these OOD tasks, our RL method maintains high robustness, achieving superior results on the diverse regional dataset. A similar trend is observed in the VisionReasoner baseline, where the RL-optimized version consistently outperforms the SFT variant, further corroborating the efficacy of RL in enhancing reasoning capabilities. This indicates that minimizing the RL objective, which directly optimizes the non-differentiable IoU, enables the model to learn more generalized geometric reasoning policies that transfer effectively across different map styles and geographic regions.

# 6 CONCLUSION

This paper introduces the task of urban socio-semantic segmentation and present SocioSeg, the first benchmark for this challenge. SocioSeg's key contribution is a new paradigm that renders heterogeneous geospatial data into a unified map, transforming a complex multi-modal problem into a visual reasoning task. We also propose SocioReasoner, a framework that leverages Vision-Language Models to mimic the human annotation process through a multi-stage reasoning segmentation workflow. By optimizing this non-differentiable pipeline with reinforcement learning, we effectively elicit the model's latent reasoning capabilities. Extensive experiments demonstrate that our approach outperforms existing methods and exhibits strong zero-shot generalization to unseen map sources. Our work highlights the potential of VLM reasoning for complex geospatial analysis.

ACKNOWLEDGMENTS

This work was supported by Amap, and in part by the National Key Research and Development Program of China under Grant 2024YFB3909001, in part by the National Natural Science Foundation of China under Grant 42371321, and in part by the Key Research and Development Program of Hubei Province under Grant 2025BAB024.

**Ethics Statement** Our research utilizes publicly accessible satellite and map data, specifically from the Amap public API, to create the SocioSeg dataset. The collection, utilization, and public re-

lease of this data, as well as our accompanying codebase, have been explicitly authorized by Amap. Consequently, both the SocioSeg dataset and the codebase are open-sourced under the Apache License 2.0. The manual annotation process was strictly confined to identifying and labeling public and private functional zones, without collecting or inferring any personally identifiable information (PII).

**Reproducibility Statement**    To ensure the full reproducibility of our findings, we have provided comprehensive implementation details throughout the paper. The construction and statistics of our SocioSeg benchmark are detailed in Sec 3 and Appendix A.1. The architecture of the SocioReasoner framework, including the multi-stage reasoning process, is described in Sec 4.1. Key details for the reinforcement learning optimization, including the reward function design and GRPO training algorithm, are presented in Sec 4.2 and Appendix A.2. In line with our commitment to open science, the SocioSeg dataset and source code will be made publicly available.

**LLM clarification**    We clarify the use of Large Language Models (LLMs) in the preparation of this manuscript. Specifically, LLMs were employed for two main purposes: translation of initial drafts from our native language and subsequent language polishing. This process involved correcting grammatical errors, improving sentence structure, and enhancing the overall readability and flow of the text. It is crucial to emphasize that all core scientific content, intellectual contributions, and original ideas presented in this paper are exclusively the work of the human authors. This includes the formulation of the research problem, the development of the SocioReasoner framework, the creation of the SocioSeg dataset, the experimental design, and the analysis of the results. The LLM served strictly as a writing aid and was not involved in any conceptual or analytical aspect of this research.

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

# A  APPENDIX

## A.1  DATASET DETAILS

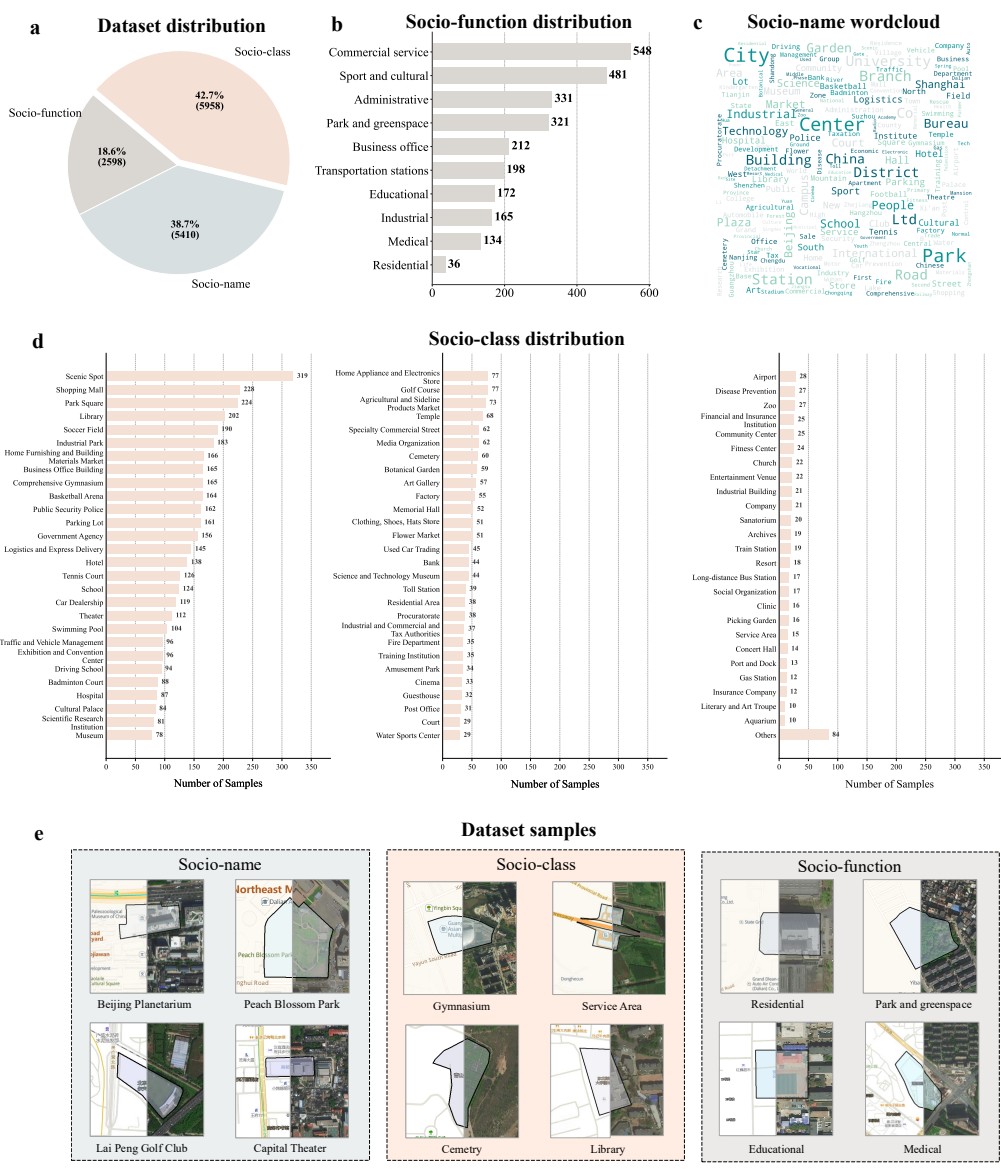

Figure 6: The SocioSeg dataset overview. (a) Sample distribution across the three hierarchical tasks. (b) Socio-function class distribution. (c) Socio-name word cloud. (d) Socio-class distribution. (e) Sample examples from SocioSeg, including satellite images, digital maps, and socio-semantic mask labels.

### A.1.1  SOCIOSEG DATASET

The SocioSeg dataset is constructed entirely from data provided by Amap, offering comprehensive geographic coverage of all provinces and major cities across China. The input modalities, namely satellite images and digital maps, are acquired via the public Amap API. The ground-truth labels are derived from Amap's Area of Interest (AOI) data. Figure 7 illustrates the data construction process.

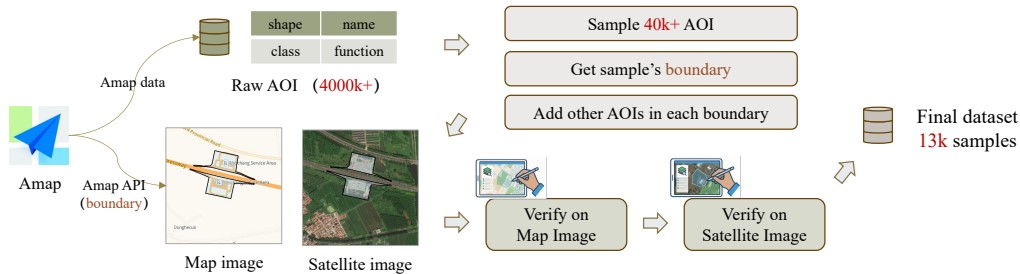

Figure 7: The SocioSeg dataset overview.

To clarify the semantic scope, we structure these labels into three hierarchical levels intrinsically linked to daily social activities: the **Socio-Name** level corresponds to the specific name of each AOI instance; the **Socio-Class** level adopts the standard third-level POI taxonomy from Amap, a classification widely utilized in prior studies (Hu & Han, 2019; Yao et al., 2017); and the **Socio-Function** level is derived from established urban functional zone definitions (Gong et al., 2020; Li et al., 2025c).

To adapt this source data for our research, we performed several refinement steps, encompassing reformatting the vector-based AOI data into rasterized semantic masks and conducting a rigorous quality assurance process. Specifically, we manually verified the alignment between the AOI polygons and the actual physical boundaries, rigorously filtering the dataset from an initial pool of approximately 40,000 samples down to 13,000 to exclude misaligned or ambiguous annotations. To further quantify the annotation quality and reproducibility, we conducted an inter-annotator agreement study with three independent annotators on a random subset of 500 samples, yielding a Cohen's Kappa coefficient of 0.854. This ensures that each pixel is precisely classified into its corresponding socio-functional category, enhancing the dataset's overall fidelity and reliability. The resulting SocioSeg benchmark is thus rich in socio-semantic information, providing a robust foundation for urban socio-semantic segmentation research.

Figure 6 offers a comprehensive overview of the SocioSeg dataset. Specifically, subfigure (a) illustrates the sample distribution across the three hierarchical tasks, underscoring the dataset's balance and diversity. Subfigures (b) and (d) present the class distributions for the socio-function and socio-class tasks, respectively, showcasing the variety of categories included. A word cloud in subfigure (c) visualizes the frequency and prominence of the socio-name labels. Finally, subfigure (e) provides qualitative examples from the dataset, displaying corresponding satellite images, digital maps, and socio-semantic masks that effectively demonstrate the data's richness and complexity.

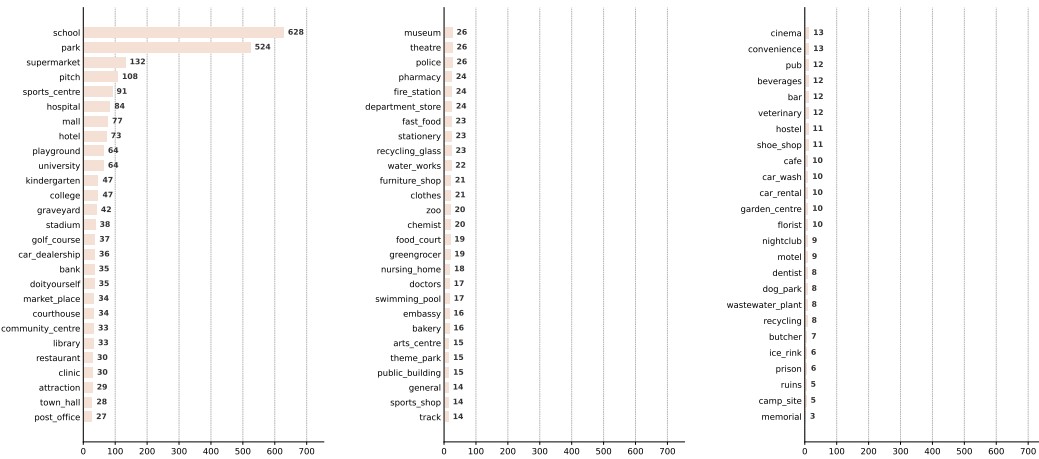

Figure 8: The SocioSeg OOD (New Region) dataset distribution.

A.1.2 SocioSeg Out-Of-Distribution dataset

To rigorously evaluate the generalization capabilities of our model, we introduce two distinct Out-Of-Distribution (OOD) datasets: **Map Style** and **New Region**.

**OOD (Map Style).** This setting is designed to assess the model's robustness to shifts in cartographic rendering and symbolization. We utilize the original test set from the SocioSeg benchmark, retaining the original satellite imagery and ground-truth labels. However, we replace the input digital map modality, originally sourced from Amap, with tiles fetched from Google Maps. This substitution isolates the impact of map style, requiring the model to generalize to a new cartographic domain without any fine-tuning.

**OOD (New Region).** Figure 8 shows the distribution of the SocioSeg OOD (New Region) dataset. This setting evaluates performance across diverse geographic environments. We constructed this dataset using OpenStreetMap (OSM) (OpenStreetMap contributors, 2017) AOI data, strictly adhering to the same data construction and quality assurance pipeline employed for the primary SocioSeg dataset. To ensure global diversity, we selected five representative cities from different continents: Tokyo (Asia), New York (North America), São Paulo (South America), London (Europe), and Nairobi (Africa). While the input digital maps are sourced from Google Maps, the ground truth labels were derived from OSM and underwent manual filtering to ensure high fidelity. The resulting dataset comprises 3,200 samples focused on the socio-class task. It encompasses 80 distinct categories, importantly including 24 novel categories that were not present in the training set, thereby posing a significant challenge for zero-shot generalization.

## A.2 Implementation Details

### A.2.1 GRPO Optimization Details

We train SocioReasoner with the two-stage end-to-end GRPO algorithm. The training process is summarized in Algorithm 1. In contrast to the single-stage training of existing methods, SocioReasoner's process includes two rounds of RL sampling and policy updates, all while utilizing a shared set of model parameters.

### A.2.2 Reward Function Design

**Format Reward Functions.** The policy generates a structured textual output $\mathbf{y}$ containing a free-form reasoning channel and a machine-parsable answer channel:

$$\langle\texttt{think}\rangle \dots \langle\texttt{/think}\rangle \quad \langle\texttt{answer}\rangle \texttt{ JSON } \langle\texttt{/answer}\rangle.$$

The answer channel must contain a valid JSON array of objects. In stage-1, each object specifies a bounding box: $\{\texttt{"bbox\_2d":} \ \texttt{[x1,y1,x2,y2]}\}$. In stage-2, each object is augmented with a list of points: $\{\texttt{"bbox\_2d":} \ \texttt{[...]}, \ \texttt{"points":} \ \texttt{[[x,y], ...]}\}$. We define a binary format reward, $R_{\text{form}}(\mathbf{y}) \in \{0, 1\}$, which is 1 if and only if the output is syntactically correct and adheres to the stage-specific schema. If the format reward is 0, the total reward for the episode is also 0, overriding all other components.

**Stage-1 (Localization) Reward.** Given the ground-truth set of boxes $\mathcal{B}^\star = \{\mathbf{b}_j^\star\}_{j=1}^J$ and the predicted set $\hat{\mathcal{B}} = \{\hat{\mathbf{b}}_k\}_{k=1}^K$, we define:

- **Format reward** $R_{\text{form}}^{(1)}(\mathbf{y})$ as defined above.
- **Accuracy reward** via Hungarian matching with an IoU threshold of 0.5. Let $\text{IoU}(\mathbf{b}, \mathbf{b}')$ be the standard box IoU. We form a binary match matrix $\mathbf{M}_{k,j} = \mathbf{1}(\text{IoU}(\hat{\mathbf{b}}_k, \mathbf{b}_j^\star) > 0.5)$, solve the linear assignment problem on the cost matrix $\mathbf{1} - \mathbf{M}$, and denote the number of matches as $N_m$. The accuracy reward is

$$R_{\text{acc}}^{(1)}(\mathbf{y}; \mathcal{B}^\star) = \frac{N_m}{\max(K, J)} \ \in \ [0, 1]. \tag{7}$$

- **Length reward** that encourages predicting the correct number of instances:

$$R_{\text{len}}^{(1)}(\mathbf{y}; \mathcal{B}^\star) = \exp\left(-2\,|K - J|/J\right), \quad J > 0. \tag{8}$$

---

**Algorithm 1** Two-stage end-to-end GRPO Training for SocioReasoner

---

**Require:** Training dataset $\mathcal{D}_{\text{train}}$; VLM policy $\pi_\theta$; frozen SAM $\mathcal{S}$; renderer $\mathcal{D}$; reference policy $\pi_{\text{ref}}$; group size $G$; PPO clip $\epsilon$; KL weight $\beta$; optimizer with learning rate $\eta$; number of RL steps $T$

1: Initialize $\pi_{\theta_{\text{old}}} \leftarrow \pi_\theta$
2: **for** step $= 1$ to $T$ **do**
3:     Sample a mini-batch $\mathcal{B} \subset \mathcal{D}_{\text{train}}$
4:     **for all** $(\mathbf{I}_s, \mathbf{I}_m, \mathbf{t}_b) \in \mathcal{B}$ **do**
5:         $\mathbf{x}_1 \leftarrow (\mathbf{I}_s, \mathbf{I}_m, \mathbf{t}_b)$                                 ▷ Stage-1: Localization
6:         **for** $g = 1$ to $G$ **do**
7:             Sample completion $\mathbf{y}_1^{(g)} \sim \pi_\theta(\cdot \,|\, \mathbf{x}_1)$
8:             Parse bounding boxes $\mathcal{B}^{(g)}$ from $\mathbf{y}_1^{(g)}$ (assign syntax reward 0 if invalid)
9:             $\mathbf{M}_c^{(g)} \leftarrow \mathcal{S}(\mathbf{I}_s, \text{prompt} = \mathcal{B}^{(g)})$
10:            Compute $R_1^{(g)}$
11:         **end for**
12:         $b_1 \leftarrow \frac{1}{G} \sum_{g=1}^{G} R_1^{(g)}$
13:         Compute advantages $A_1^{(g)} \leftarrow R_1^{(g)} - b_1$ for all $g$
14:         Update policy $\pi_\theta$ with GRPO on $\{\mathbf{x}_1, \mathbf{y}_1^{(g)}, A_1^{(g)}\}_{g=1}^{G}$, using clip $\epsilon$ and KL weight $\beta$
15:         Select $g^\star \leftarrow \arg\max_g R_1^{(g)}$ (or sample proportional to $\exp(R_1^{(g)})$)
16:         $\mathbf{I}_{s,r} \leftarrow \mathcal{D}(\mathbf{I}_s, \mathcal{B}^{(g^\star)}, \mathbf{M}_c^{(g^\star)})$
17:         $\mathbf{I}_{m,r} \leftarrow \mathcal{D}(\mathbf{I}_m, \mathcal{B}^{(g^\star)}, \mathbf{M}_c^{(g^\star)})$
18:         $\mathbf{x}_2 \leftarrow (\mathbf{I}_{s,r}, \mathbf{I}_{m,r}, \mathbf{t}_b, \mathbf{M}_c^{(g^\star)})$                ▷ Stage-2: Refinement
19:         **for** $g = 1$ to $G$ **do**
20:             Sample completion $\mathbf{y}_2^{(g)} \sim \pi_\theta(\cdot \,|\, \mathbf{x}_2)$
21:             Parse $\{\tilde{\mathcal{B}}^{(g)}, \mathcal{P}^{(g)}\}$ from $\mathbf{y}_2^{(g)}$ (assign syntax reward 0 if invalid)
22:             $\mathbf{M}_f^{(g)} \leftarrow \mathcal{S}(\mathbf{I}_s, \text{prompt} = \{\tilde{\mathcal{B}}^{(g)}, \mathcal{P}^{(g)}\})$
23:            Compute $R_2^{(g)}$
24:         **end for**
25:         $b_2 \leftarrow \frac{1}{G} \sum_{g=1}^{G} R_2^{(g)}$
26:         Compute advantages $A_2^{(g)} \leftarrow R_2^{(g)} - b_2$ for all $g$
27:         Update policy $\pi_\theta$ with GRPO on $\{\mathbf{x}_2, \mathbf{y}_2^{(g)}, A_2^{(g)}\}_{g=1}^{G}$, using clip $\epsilon$ and KL weight $\beta$
28:     **end for**
29:     $\pi_{\theta_{\text{old}}} \leftarrow \pi_\theta$                                         ▷ Refresh behavior policy for next step
30: **end for**

---

The total stage-1 reward is the unweighted sum of these components:

$$R_1(\mathbf{y}; \mathbf{x}) = R_{\text{form}}^{(1)}(\mathbf{y}) + R_{\text{acc}}^{(1)}(\mathbf{y}; \mathcal{B}^\star) + R_{\text{len}}^{(1)}(\mathbf{y}; \mathcal{B}^\star). \tag{9}$$

**Stage-2 (Refinement) Reward.** For each predicted group (one bbox plus its point list), we execute SAM with the prompts to obtain a mask $\hat{\mathbf{M}}_f$ and compare it to the ground-truth mask $\mathbf{M}^\star$:

- **Format reward** $R_{\text{form}}^{(2)}(\mathbf{y})$ as defined above.
- **Accuracy reward** as pixel IoU:

$$R_{\text{acc}}^{(2)}(\mathbf{y}; \mathbf{x}) = \text{IoU}(\hat{\mathbf{M}}_f, \mathbf{M}^\star) \in [0, 1]. \tag{10}$$

- **Length reward** that encourages concise, informative interactions. For a group with $n$ points, we define a Gaussian-shaped score peaking at two points:

$$R_{\text{len}}^{(2)}(\mathbf{y}) = \frac{1}{G'} \sum_{g=1}^{G'} r(n) \in [0, 1], \tag{11}$$

Table 5: All ablation of multi-stage.

| Method | Socio-name | | | Socio-class | | | Socio-function | | | All dataset | | |
|---|---|---|---|---|---|---|---|---|---|---|---|---|
| | cIoU | gIoU | F1 | cIoU | gIoU | F1 | cIoU | gIoU | F1 | cIoU | gIoU | F1 |
| w/o reflection | 48.5 | 50.9 | 58.4 | 44.4 | 49.3 | 55.5 | 36.3 | 41.8 | 45.0 | 44.0 | 48.5 | 54.3 |
| w/o refinement | 50.5 | 53.1 | 61.2 | 46.2 | 51.0 | 58.1 | 40.3 | 45.7 | 48.1 | 46.4 | 50.8 | 57.5 |
| Ours | **52.6** | **55.7** | **64.6** | **47.6** | **52.8** | **60.1** | **40.6** | **46.9** | **50.3** | **47.9** | **52.8** | **59.7** |

Table 6: All ablation of point number.

| Method | Socio-name | | | Socio-class | | | Socio-function | | | All dataset | | |
|---|---|---|---|---|---|---|---|---|---|---|---|---|
| | cIoU | gIoU | F1 | cIoU | gIoU | F1 | cIoU | gIoU | F1 | cIoU | gIoU | F1 |
| 1 point refinement | 51.6 | 53.4 | 61.2 | 47.6 | 51.2 | 59.0 | 40.0 | 45.7 | 49.5 | 47.6 | 51.2 | 58.0 |
| 2 points refinement | 52.6 | **55.7** | 64.6 | 47.6 | **52.8** | **60.1** | 40.6 | **46.9** | **50.3** | 47.9 | **52.8** | **59.7** |
| 3 points refinement | **53.2** | 54.7 | **65.0** | **48.9** | 52.6 | 59.7 | **41.8** | 46.6 | 49.8 | **48.9** | 52.3 | 58.8 |

where $r(n) = \exp\left(-\frac{(n-\mu)^2}{2\sigma^2}\right)$ with $\mu = 2$ and $\sigma = 2$. $G'$ is the number of valid groups. This encourages using a small number of informative points rather than many redundant ones.

The total stage-2 reward is the sum of these components:

$$R_2(\mathbf{y}; \mathbf{x}) = R_{\text{form}}^{(2)}(\mathbf{y}) + R_{\text{acc}}^{(2)}(\mathbf{y}; \mathbf{x}) + R_{\text{len}}^{(2)}(\mathbf{y}). \tag{12}$$

### A.2.3 EXPERIMENTAL SETTINGS

For all our Reinforcement Learning (RL) based models, namely VisionReasoner, Seg-R1, SAM-R1, and RemoteReasoner, we adopt a unified training configuration. We set the rollout batch size to 128 and the group size to 8. The models are optimized using the AdamW optimizer with a learning rate of $1 \times 10^{-6}$. For the Proximal Policy Optimization (PPO) algorithm, the clipping parameter $\epsilon$ is set to 0.5, and the Kullback-Leibler (KL) divergence weight $\beta$ is configured to 0.005. All RL models are trained for 250 steps within the ROLL framework (Wang et al., 2025).

A key aspect of our methodology is the handling of visual inputs. Since all RL-based methods are built upon Qwen2.5-VL-3b, which natively supports multi-image inputs, we provide both the satellite imagery and digital maps as visual input. For the Supervised Fine-Tuning (SFT) version of our model, we construct the supervision signal using the bounding box of the ground-truth mask, along with three points randomly sampled from within the mask's area.

In contrast, for the baseline models UNet, SegFormer, SegEarth-OV, RSRefSeg and SegEarth-R1, we followed the original authors' implementations. We utilized their publicly available source code and pre-trained models, which are then fine-tuned on the SocioSeg dataset. As these architectures do not support multi-image inputs, only the satellite imagery is used as the visual input for these models. All models are trained on a high-performance computing cluster equipped with 16 NVIDIA H20 GPUs.

### A.3 USER PROMPT TEMPLATE

The user prompt templates utilized in our experiments are shown in Figure 9. SocioReasoner employs a two-stage reasoning process; consequently, we designed two distinct prompt templates to accommodate the different input and output formats of each stage. For our baseline model without the reflection mechanism, as well as the GPT and Qwen models, we adopt a single-stage prompt template. This template is adapted from the one used by VisionReasoner, with modifications to meet the specific requirements of our task. For our SFT model, we use this same base template but remove the chain-of-thought components. For all other RL-based comparative methods, we used the original prompt templates provided by their respective authors, prepending each with the instruction, "You will be given two images. The first is a map and the second is a corresponding satellite image."

**Stage-1 User Prompt**

"You will be given two images. The first is a map and the second is a corresponding satellite image."
"Please find '{Query}' with bboxs."
"Compare the difference between object(s) and find the most closely matched object(s)."
"Output the thinking process in <think> </think> and final answer in <answer> </answer> tags. "
"Please use English. Output the bbox(es) in JSON format."

"i.e., <think>thinking process here </think>"
"<answer>[{"bbox_2d": [bx1,by1,bx2,by2]}, {"bbox_2d": [bx3,by3,bx4,by4]}]</answer>"

**Stage-2 User Prompt**

"You will be given two images. The first is a map and the second is a corresponding satellite image."
"Now some bbox(s) and the results after SAM segmentation for '{Query}' have been rendered on these two images."
"The found bbox(s) are: {Bboxs}."
"Please add some points appropriately to each bbox to better represent the area of interest."
"Output the thinking process in <think> </think> and final answer in <answer> </answer> tags."

"i.e., <think> thinking process here </think>"
"<answer>[{"bbox_2d": [bx1,by1,bx2,by2], "points": [[px1,py1],[px2,py2],[px3,py3]]}]</answer>"

**Single-stage User Prompt**

"You will be given two images. The first is a map and the second is a corresponding satellite image."
"Please find '{Query}' with bboxs and some points appropriately to each bbox to better represent the area of interest. "
"Compare the difference between object(s) and find the most closely matched object(s)."
"Output the thinking process in <think> </think> and final answer in <answer> </answer> tags. "
"Please use English. Output the bbox(es) in JSON format."

"i.e., <think>thinking process here </think>"
"<answer>[{"bbox_2d": [bx1,by1,bx2,by2], "points": [[px1,py1],[px2,py2],[px3,py3]]}]</answer>"

Figure 9: The two prompts above are the user prompt template for SocioReasoner, which adopts a two-stage reasoning process to mimic human annotation. The prompt below is the single-stage prompt used for the baseline without reflection and zero-shot GPT and Qwen models.

Table 7: Comparison with state-of-the-art methods on the SocioSeg OOD (New Region) dataset.

| Method | OOD (New Region) Dataset | | |
| --- | --- | --- | --- |
| | cIoU | gIoU | F1 |
| UNet | 10.0 | 7.3 | 7.1 |
| Segformer | 18.8 | 14.7 | 14.1 |
| VisionReasoner | 32.8 | 34.4 | 35.0 |
| Seg-R1 | 28.8 | 28.8 | 26.2 |
| SAM-R1 | 14.8 | 14.5 | 19.0 |
| SegEarth-OV | 3.2 | 3.2 | 0.0 |
| RSRefSeg | 12.4 | 10.3 | 13.8 |
| SegEarth-R1 | 20.7 | 18.7 | 21.2 |
| RemoteReasoner | 27.5 | 29.8 | 28.1 |
| **Ours** | **40.2** | **43.4** | **42.9** |

## A.4 SocioSeg benchmark

GPT-5, GPT-o3 and Qwen2.5-VL-72b are evaluated as baselines without any fine-tuning. As shown in Table 9, their performance is substantially lower than that of our trained model, indicating that even large-scale VLMs struggle with the complexities of socio-semantic segmentation without task-specific training. Notably, Qwen2.5-VL-3b fails to produce valid bounding box outputs in our experiments, resulting in zero performance. This underscores the importance of specialized training and the effectiveness of our reinforcement learning approach in eliciting the reasoning capabilities necessary for this task.

Table 8: Inference time comparison (seconds per sample).

| VisionReasoner | SegR1 | SAMR1 | RSRefSeg | SegEarth-R1 | RemoteReasoner | Ours-rl | Ours-sft |
|---|---|---|---|---|---|---|---|
| 1.33 | 1.07 | 2.52 | 0.16 | 0.35 | 1.13 | 2.71 | 0.41 |

Table 9: SocioSeg benchmark.

| Method | Backbone | Socio-name | | Socio-class | | Socio-function | | All dataset | |
|---|---|---|---|---|---|---|---|---|---|
| | | cIoU | gIoU | cIoU | gIoU | cIoU | gIoU | cIoU | gIoU |
| GPT-5 | Not disclosed | 16.1 | 16.1 | 14.9 | 15.1 | 12.2 | 12.5 | 14.7 | 15.0 |
| GPT-o3 | Not disclosed | 22.6 | 22.9 | 20.9 | 22.7 | 16.1 | 17.3 | 20.3 | 21.7 |
| Qwen2.5-VL-3b | Qwen2.5-VL-3b | 0.0 | 0.0 | 0.0 | 0.0 | 0.0 | 0.0 | 0.0 | 0.0 |
| Qwen2.5-VL-72b | Qwen2.5-VL-72b | 27.1 | 29.5 | 21.8 | 27.2 | 20.4 | 24.4 | 23.1 | 27.5 |

## A.5 MORE QUANTITATIVE RESULTS

### A.5.1 MORE ABLATION STUDIES AND INFERENCE TIME COMPARISON

We provide the comprehensive quantitative results of our ablation studies in Table 5 and Table 6, presenting detailed metrics across all three hierarchical task levels. Furthermore, Table 5 reports the performance of all comparative methods on the OOD (New Region) dataset, demonstrating that our method consistently achieves the best results across all evaluation metrics. Finally, we present a computational efficiency analysis in Table 8, comparing the average inference time per sample (in seconds) against the baselines. While our model incurs a higher latency due to its iterative two-stage reasoning mechanism, this computational trade-off is justified by its superior segmentation accuracy.

## A.6 MORE VISUALIZATIONS

We first present the trend of the reward function during the training process, as shown in Figure 10. As can be seen, the reward function gradually converges as training progresses, indicating that the model continuously improves its decision-making quality. Next, we provide additional qualitative results comparing SocioReasoner's performance on the three hierarchical tasks, as illustrated in Figure 11. These examples clearly demonstrate the advantages of SocioReasoner's performance across different tasks, especially its accuracy and robustness in complex scenarios. Furthermore, we showcase more inference examples from SocioReasoner in Figure 13. These examples further validate SocioReasoner's capability in processing multi-modal inputs.

## A.7 FAILURE CASES

Finally, we present a visualization of representative failure cases encountered by SocioReasoner in Figure 12. These instances, all characterized by a Generalized IoU (gIoU) below 0.1, primarily stem from two distinct error modes: (i) Localization Failure (Cases 1 and 2), where the predicted bounding box fails to locate the target region entirely, likely due to the model being distracted by the dense and visually cluttered urban environment. In such cases, the high density and low resolution of the Points of Interest rendered on the map make it difficult for the VLM to discern the correct locations of the semantic entities; and (ii) Boundary Imprecision (Cases 3, 4, and 5), where the model correctly identifies the general location of the semantic entity but fails to generate a geometrically accurate enclosure. This issue can be attributed to the complexity of geographical features in satellite imagery, where the boundaries of buildings, roads, and other urban elements are often ambiguous and irregular, making it difficult for the VLM to infer precise bounding boxes. Furthermore, certain semantic categories inherently possess highly abstract definitions, whose physical boundaries may not be clear-cut or fixed, which further compounds the difficulty of accurate localization. These challenges underscore the inherent difficulty of mapping abstract social concepts to precise physical coordinates in complex satellite imagery. This suggests that while our reasoning framework effectively bridges the modal gap, future research should further focus on enhancing the fine-grained spatial reasoning capabilities of Vision-Language Models (VLMs) and mitigating spatial ambiguity to achieve more robust performance in dense urban scenarios.

Table 10: Multi-class semantic segmentation setting.

| Method | Socio-class | | | Socio-function | | |
|---|---|---|---|---|---|---|
| | cIoU | gIoU | F1 | cIoU | gIoU | F1 |
| UNet | 0.1 | 0.1 | 0.1 | 0.7 | 0.4 | 1.3 |
| SegFormer | 8.4 | 9.5 | 15.5 | 4.2 | 4.2 | 8.0 |

### A.8 APPLICATION

In the industrial sector, the practical value of our method lies in its ability to infer the corresponding spatial extent, specifically the Area of Interest (AOI), from a given Point of Interest (POI). This provides essential visualization data for mapping applications and is of significant importance for coordinate-based recommendation systems. For instance, deriving precise AOIs enables superior Location-Based Services (LBS) (Huang et al., 2018) recommendations, such as for dining, retail, and entertainment venues, thereby enhancing overall user experience and satisfaction. Furthermore, accurate AOIs allow mapping applications to optimize route planning and navigation, providing more reliable route suggestions and spatial context, especially within complex urban environments.

In academic contexts, particularly for paradigms like the 15-minute city, precise AOIs assist urban planners and policymakers in comprehensively understanding urban spatial structures and functional distributions, which in turn optimizes public resource allocation and elevates the quality of urban life. For example, current 15-minute city frameworks (Bruno et al., 2024) are predominantly calculated using POIs, whose spatial boundaries are fundamentally ambiguous. By acquiring precise AOIs, it becomes possible to rigorously evaluate actual spatial accessibility and service coverage, thereby providing a more scientific foundation for data-driven urban planning.

### A.9 LIMITATIONS AND FUTURE WORK

While the SocioSeg dataset provides a robust benchmark for urban socio-semantic segmentation, we acknowledge a limitation regarding the handling of multi-instance targets. In real-world scenarios, a single satellite image may contain multiple instances of the same Socio-class or Socio-function (e.g., multiple office buildings). In our dataset, the ratio of single-instance to multi-instance samples is 0.89:0.11, with an average of 1.17 instances per image.

During our experiments, we observed that SocioReasoner, along with other VLM+SAM based baselines (such as VisionReasoner and RemoteReasoner), tends to converge toward identifying and segmenting a single dominant instance. This behavior reflects a known bottleneck in the spatial reasoning and multi-target localization capabilities of current Vision-Language Models. It is important to note, however, that our comparative evaluations remain strictly fair, as all methods were trained and evaluated under identical conditions and constraints on this dataset.

In future work, we aim to address this single-instance bias. Recent studies suggest that scaling up the parameter size of the base VLM (e.g., 7B parameters or larger) can significantly enhance multi-instance perception and fine-grained spatial reasoning. Integrating larger models or developing specialized multi-instance generation constraints within the reinforcement learning reward function represents a promising direction to further improve socio-semantic segmentation performance in dense urban environments.

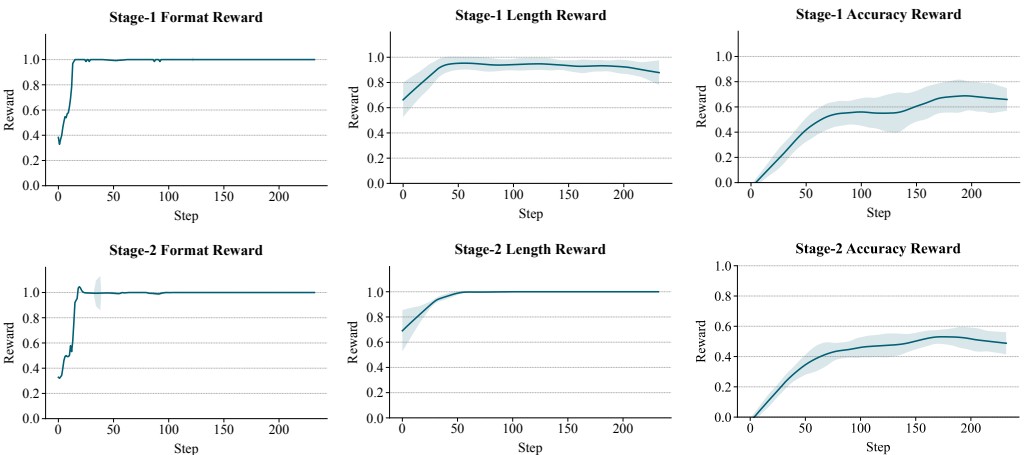

Figure 10: The rewards visualization during the training process.

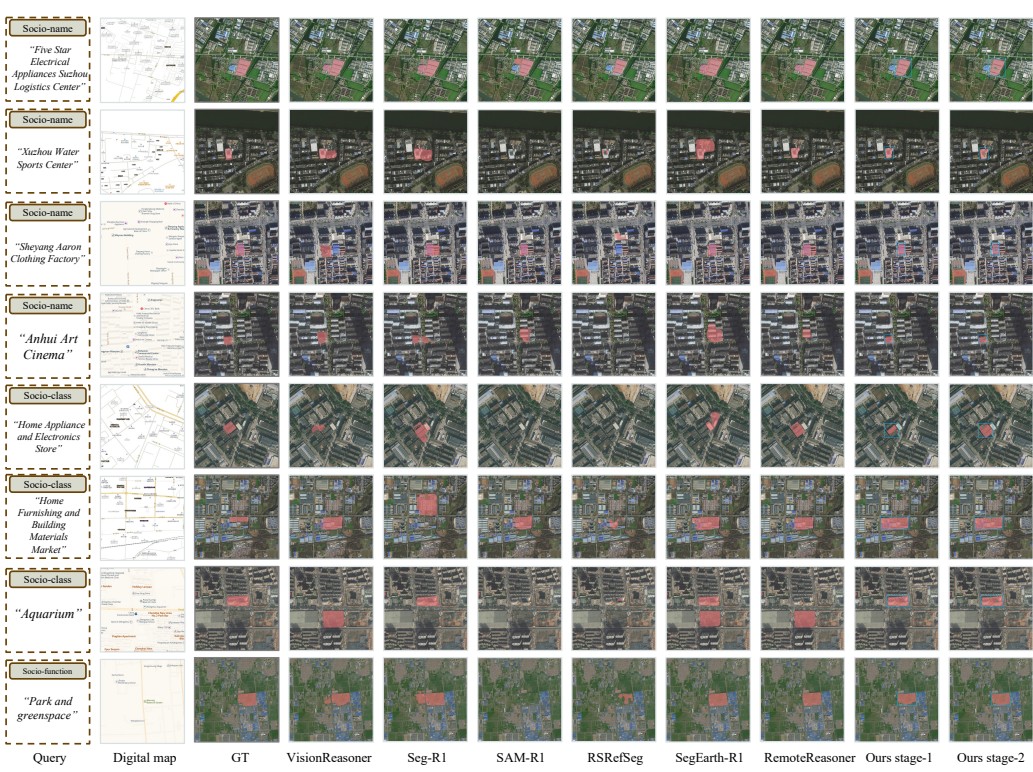

Figure 11: All method Comparisons of SocioReasoner across the three hierarchical tasks.

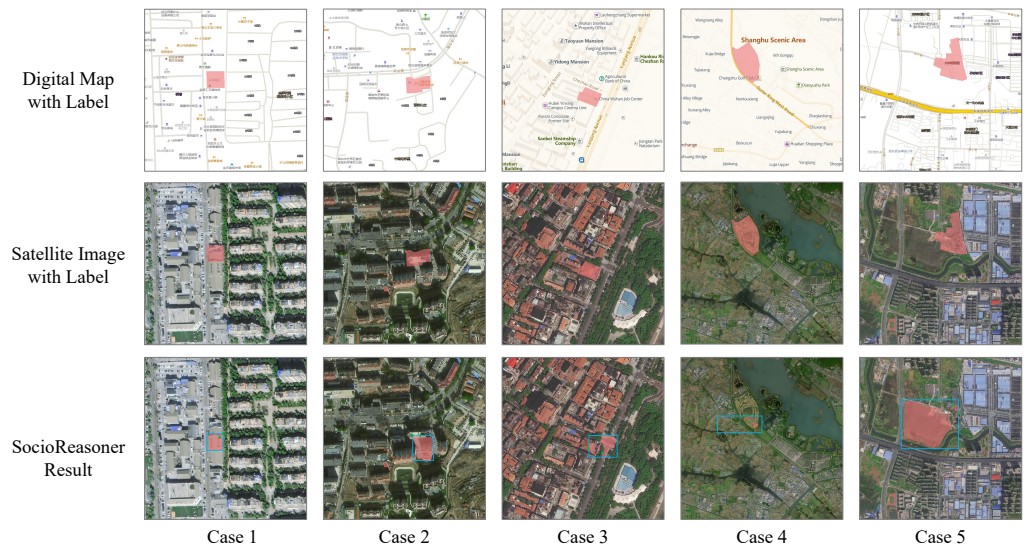

Figure 12: Failure cases of SocioReasoner.

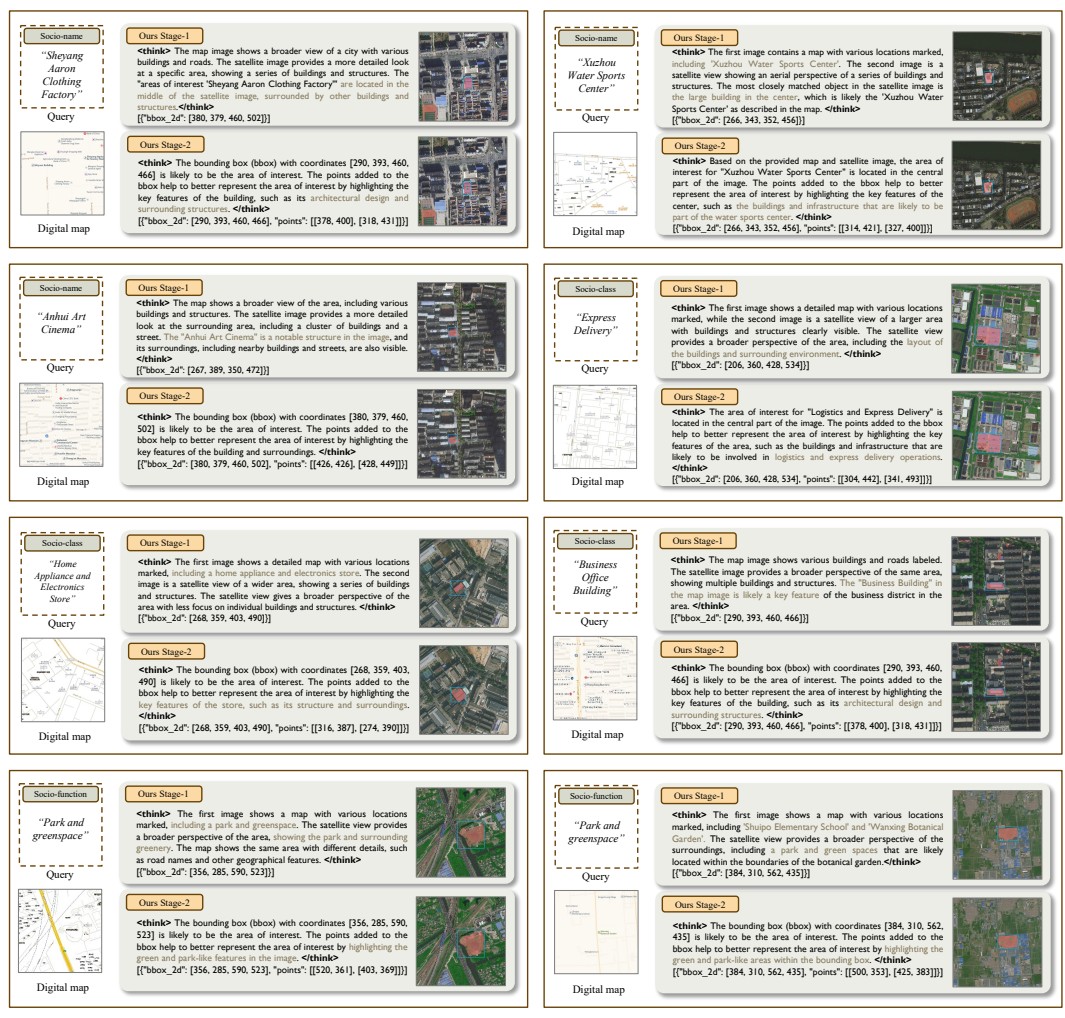

Figure 13: More inference examples of SocioReasoner.

