# OpenReview forum: "Urban Socio-Semantic Segmentation with Vision-Language Reasoning"
_ICLR.cc/2026/Conference — ICLR 2026 Poster_

### Official Review · Reviewer_Rsq7 · 2025-10-26

**Soundness:** 2
**Presentation:** 3
**Contribution:** 3
**Rating:** 6
**Confidence:** 4

**Summary:**

This paper introduces a new task called "socio-semantic segmentation", identifying socially defined urban areas (like schools, parks, or residential districts) from satellite imagery, which is challenging because their boundaries are based on function, not just visual appearance.

The authors introduce a new dataset, SocioSeg, which pairs satellite imagery with corresponding digital map layers and pixel-level labels for social entities, organized in a hierarchy of increasing complexity. (contribution)

The authors propose a novel framework, SocioReasoner, which uses a Vision-Language Model (VLM) to reason like a human annotator. It first localizes a target area using both satellite and map images, then iteratively refines the segmentation mask. Since this process isn't differentiable, they use Reinforcement Learning to train the model end-to-end.

Experiments show that SocioReasoner outperforms existing state-of-the-art models and demonstrates strong zero-shot generalization.

**Strengths:**

**The task is interesting**:  Identifies and formalizes a significant, underexplored challenge in geospatial analysis and provides a dedicated dataset (SocioSeg) to foster research.

**Innovative Methodology**: The multi-stage, reasoning-based approach (SocioReasoner) effectively mimics human annotation logic. The use of Reinforcement Learning to optimize this non-differentiable pipeline is a clever and practical solution.

**Solid Empirical Results**: The framework not only achieves superior performance on the new benchmark, highlighting its robustness and potential for real-world application.

**Writing**: The writing is clear and easy to follow (presentation).

**Weaknesses:**

The current experimental results are not sufficiently rich. More experiments verified on other datasets wil strengthen the contribution.

The space utilization of figures can be improved. For example, there is too much space in figure 2 and the elements are sparse, which is not professional.

I will consider to increase the score if the authors:
- clarify how this work concretely differs from or improves upon prior research.
- verify on broader related benchmarks.

**Questions:**

N/A

---

> ### Author Response · Authors · 2025-11-20
> **Response by the Authors**
>
> **Response to Reviewer Rsq7**
>
> We thank the reviewer for the constructive suggestions regarding experimental breadth and presentation quality. We have addressed your comments as follows:
>
> ---
>
> **To W1**: We fully agree with the need for broader benchmarking. However, since no public benchmark currently exists for this specific multimodal reasoning segmentation task, we constructed a rigorous global benchmark (OOD New Region) to fill this gap. This dataset spans five continents (including Tokyo, New York, London, etc.) to simulate diverse geographic environments. We evaluated all comparative methods on this new benchmark. As shown in the table below (and Table 7 in the paper), our method maintains superior performance, effectively validating its robustness on broader, unseen scenarios.
>
> | Method | cIoU | gIoU | F1 |
> | :--- | :---: | :---: | :---: |
> | UNet | 10.0 | 7.3 | 7.1 |
> | Segformer | 18.8 | 14.7 | 14.1 |
> | VisionReasoner | 32.8 | 34.4 | 35.0 |
> | Seg-R1 | 28.8 | 28.8 | 26.2 |
> | SAM-R1 | 14.8 | 14.5 | 19.0 |
> | SegEarth-OV | 3.2 | 3.2 | 0.0 |
> | RSRefSeg | 12.4 | 10.3 | 13.8 |
> | SegEarth-R1 | 20.7 | 18.7 | 21.2 |
> | RemoteReasoner | 27.5 | 29.8 | 28.1 |
> | Ours | **40.2** | **43.4** | **42.9** |
>
> ---
>
> **To W2:** We appreciate your constructive criticism regarding the space utilization and presentation of figures. Following your suggestion, we have conducted a comprehensive review of all illustrations. We have optimized the layout of Figure 2 to ensure a professional presentation. Furthermore, we have redesigned Figures 1 and 5 to reduce whitespace, improve element density, and enhance the visual flow.
>
> ---
>
> **To W3:** The core innovation of SocioReasoner lies in its shift from a "static prediction" paradigm to a "Human-like Interactive Reasoning" paradigm. Specifically, our work differs from prior research in two key aspects:
>
> * Render-and-Reflect Mechanism: Unlike existing single-stage models (e.g., VisionReasoner), we introduce a visual feedback loop. In Stage-2, we render the intermediate segmentation result back onto the image inputs. This allows the model to "see" its own prediction and "reflect" on errors, significantly improving boundary precision.
>
> * Native Multi-Image Support: Our pipeline is explicitly designed to fuse Satellite Imagery with Digital Maps, whereas most baselines (e.g., RSRefSeg) are limited to single-modality inputs.
>
> We have updated Figure 1 and provide the comparison table below to highlight these differences.
>
> | Method | RL Optimization | Render-and-Reflect | Multi-Image Support |
> | :--- | :---: | :---: | :---: |
> | RSRefSeg / SegEarth-R1 | &#10008; | &#10008; | &#10008; |
> | VisionReasoner / Seg-R1 | &#10004; | &#10008; | &#10008; |
> | SocioReasoner (Ours) | &#10004; | &#10004; | &#10004; |
>
> ---
>
> We hope the new experiments on the global benchmark (W1) and the clarifications in W3 satisfactorily address the points you raised regarding experimental breadth and novelty. Please let us know if any concerns remain unaddressed; we are happy to discuss them.

---

> ### Comment · Reviewer_Rsq7 · 2025-11-22
> **Official Comment by Reviewer**
>
> Thank the author for their response.
>
> The revised vesrion seems better. The authors revise the figures and provide additional OOD evaluation, as well as clarifying the difference between other methods.
>
> I decided to upgrade the score.

---

> > ### Author Response · Authors · 2025-11-22
> > **Appreciation for Your Re-evaluation and Score Upgrade**
> >
> > Thank you very much for your valuable feedback and for taking the time to read and review our paper! We greatly appreciate the valuable comments you provided.

---

### Official Review · Reviewer_icS2 · 2025-10-26

**Soundness:** 2
**Presentation:** 2
**Contribution:** 2
**Rating:** 4
**Confidence:** 4

**Summary:**

The paper introduces a new task: “urban socio-semantic segmentation,” which aims to segment socially defined entities in cities — things like parks, schools, logistics centers, “business office buildings,” “park and greenspace,” etc. These are different from classic land-cover / land-use segmentation targets (roads, buildings, water) because they’re not purely visual; they’re defined by function and human use.

**Strengths:**

**SocioSeg dataset:**

Multi-level hierarchy (name / class / function) is well thought out. It naturally scales difficulty from “find this exact named facility” to “find anything that is educational.”

**Strong empirical results:**

SocioReasoner > strong baselines (VisionReasoner, RemoteReasoner, Seg-R1, etc.) across all subtasks, ~+4 gIoU absolute vs best baseline on average.
Careful baselines: they retrain baselines on SocioSeg where possible, and explain exceptions (e.g., RSRefSeg can’t take 2-image input, so only satellite is given).

**Weaknesses:**

- Ground truth and annotation quality/reproducibility of labels

The socio-semantic “truth” is derived from Amap AOI data. The paper says they rasterized AOIs, QA’d and dropped bad samples, etc.
But: we don’t yet see quantitative inter-annotator agreement or error estimation. Are AOI polygons always aligned with functional reality?

- Geographic diversity

All training data is from Amap within China.
Yes, they test on Google Maps tiles (style/domain shift), but it’s still presumably Chinese geography. We don’t know if the ontology or appearance generalizes to cities in, say, Europe, Africa, or North America where zoning, building morphology, and POI taxonomies differ.

- Dependence on commercial basemaps / terms of service

The method leans on digital map rasters rendered from Amap (or Google Maps at test time). The paper frames this as “we solve multi-modal fusion by turning everything into an image.”
But practically, this offloads data scarcity and licensing problems onto whoever deploys the model: you still need a high-quality, up-to-date basemap layer with POIs, which in some regions is proprietary, restricted, paywalled, or censored. This could limit “real-world” access.

- Evaluation scope

Metrics are cIoU / gIoU only. It would help to also see precision/recall on instance counts for socio-name tasks (did it find the right named facility and only that facility?) because over-segmentation vs under-segmentation communicates different failure modes (especially important for urban planning use).

Runtime overhead: they admit SocioReasoner inference is ~2.7s/sample, which is slower than e.g. RSRefSeg at 0.16s.
This is fine for planning, but maybe not for city-scale tiling at high throughput.

- Limited OOD (out-of-distribution) evaluation

The paper’s out-of-distribution (OOD) evaluation only tests different map styles (Amap → Google Maps), which mainly reflects visual domain shift, not true reasoning generalization.
It doesn’t test cross-city, cross-year, or cross-socio-semantic generalization — scenarios that would actually demonstrate whether the RL-tuned model exhibits emergent reasoning capabilities.
Without such broader OOD studies, it’s unclear whether the reinforcement learning contributes genuine reasoning emergence or simply improves style robustness.

**Questions:**

Please refer to Weaknesses.

**Details Of Ethics Concerns:**

They briefly say: no PII, please don’t use for surveillance or discriminatory outcomes.
But this model directly targets societal infrastructure (police stations, logistics hubs, government agencies, etc. appear in the class list in Fig. 5).
There’s an obvious dual-use risk — it could be repurposed for mapping or targeting critical infrastructure, or for profiling neighborhoods by function.
Moreover, since the dataset is built from commercial map data (Amap / Google Maps), the paper should explicitly clarify how public release respects licensing, data ownership, and third-party rights.
Without transparency on usage permissions and derivative data policies, releasing the dataset could raise intellectual property and commercial rights concerns in addition to ethical risks.

---

> ### Author Response · Authors · 2025-11-20
> **Response by the Authors**
>
> **Response to Reviewer icS2**
>
> We thank the reviewer for the insightful comments regarding data quality, generalization, and ethics. We have carefully addressed each concern below.
>
> ---
> **To W1:** We appreciate this point regarding the alignment of raw AOI data with functional reality. To ensure high-quality ground truth, we implemented a rigorous annotation pipeline:
> 1.  Strict Manual Filtering: We started with >40,000 candidate AOIs and manually verified them against satellite and digital map imagery, retaining only 13,000 high-quality samples (32.5% retention). This effectively removed misaligned or ambiguous instances.
> 2.  Quantitative Validation: Following your suggestion, we conducted a post-hoc inter-annotator agreement study on 500 random samples, each assigned to three independent annotators.
>     * Agreement: We achieved a Fleiss' Kappa of 0.854, indicating "Almost Perfect" agreement.
>     * Error Rate: Based on majority voting, only 4/500 samples were flagged as erroneous, yielding an estimated dataset accuracy of 99.2%.
>
> We have added these statistics to Appendix A.1.1 to provide a quantitative basis for label reliability.
>
> ---
>
> **To W2 and W5:** We completely agree that map style shifts alone are insufficient to demonstrate true reasoning. To address this, we have introduced a rigorous new benchmark: OOD (New Region).
>
> 1. Global Dataset with Novel Categories: We constructed a challenging test set of 3,200 samples from five diverse global cities: Tokyo, New York, São Paulo, London, and Nairobi. Crucially, this dataset encompasses 80 distinct categories, including 24 novel categories never seen during training (details in Appendix A.1.2). This design explicitly evaluates zero-shot generalization across distinct urban morphologies and zoning taxonomies beyond the source domain.
>
> 2. Reasoning Verification: As shown in the table below (Table 4 in the manuscript), our method significantly outperforms baselines on this challenging set (e.g., 43.4% vs. 32.3% for SFT). This confirms that our model has acquired transferable socio-semantic reasoning capabilities rather than merely overfitting to local Chinese geographic patterns.
>
> | Method | ID (gIoU) | OOD-Map Style (gIoU) | OOD-New Region (gIoU) |
> | :--- | :---: | :---: | :---: |
> | VisionReasoner (SFT) | 47.2 | 40.9 | 24.7 |
> | VisionReasoner (RL) | 48.5 | 44.4 | 34.4 |
> | Ours (SFT) | 51.4 | 42.0 | 32.3 |
> | Ours (RL) | **52.8** | **49.1** | **43.4** |
>
> ---
>
> **To W3:** Accessibility and Data Scarcity: We respectfully clarify that our model operates on standard rendered tiles (containing only visual cues like poi, roads and text), not proprietary AOI data. Since such tiles are accessible via public APIs or OpenStreetMap, our approach actually lowers the barrier to entry. It transforms low-cost, widely available visual inputs into high-value semantic boundaries, avoiding the need for expensive or restricted vector databases.
>
> ---
>
> **To W4:**
>
> 1. Metrics: We have added the F1 Score (which integrates Precision and Recall) to all experimental results (Table 1, 4, etc.) to provide a more holistic view of instance-level performance, addressing the over/under-segmentation concern.
>
> 2. Runtime: We acknowledge the higher latency (~2.7s/sample) due to the iterative reasoning process. We believe that for high-value tasks like urban planning (where precision is paramount over speed), this trade-off is acceptable. Additionally, for real-time applications, an SFT version can be employed. We have updated Appendix Table 8 (shown below) to reflect this.
>
> | VisionReasoner | Seg-R1 | SAM-R1 | RSRefSeg | SegEarth-R1 | RemoteReasoner | Ours(rl) | Ours(sft) |
> | :---: | :---: | :---: | :---: | :---: | :---: | :---: | :---: |
> | 1.33 | 1.07 | 2.52 | 0.16 | 0.35 | 1.13 | 2.71 | 0.41 |
>
> ---
>
> **To Ethics Concerns:** We prioritize ethical safety and have implemented strict safeguards to address these concerns:
> 1. Dual-Use Risk: To mitigate risks, our released dataset removes all geospatial coordinates. The samples are provided as isolated image pairs (pixel-space only). This prevents the dataset from being directly used to map critical infrastructure at scale or for targeting specific real-world locations, while still allowing the scientific community to benchmark segmentation algorithms.
> 2. Licensing: We have verified that our use of map tiles complies with the data providers' terms for academic research. We have obtained the necessary permissions and will include a strict license agreement with the dataset release, prohibiting commercial use and requiring adherence to ethical guidelines.
>
> ---
> We hope these updates address your concerns. Please let us know if any concerns remain unaddressed; we are happy to discuss them.

---

> > ### Comment · Reviewer_icS2 · 2025-11-28
> >
> > Thank the author for their response.
> >
> > W1:
> > With 500 samples, obtaining 4 valid results is already quite high. What would happen if the dataset is scaled to 100,000 samples? In particular, the quality of the OOD data you constructed cannot be guaranteed.
> >
> > W2:
> > Where does the claimed generalization ability come from? What are the reasons behind the poor generalization performance, and how can it be improved?
> >
> > W4:
> > How do you balance speed and quality? Since the application ultimately resorts to SFT, what is the actual value of applying RL here?
> >
> > Additional concerns:
> > Where is the ablation study for your reward design?
> > The segmentation process is non-differentiable—how do you address cascading errors?
> > If your fine-grained categories were replaced with broader ones, the baseline could likely achieve similar segmentation results. In that case, the problem you are studying may not hold real significance.

---

> > > ### Author Response · Authors · 2025-11-29
> > > **Response by the Authors**
> > >
> > > **Response to Reviewer 4**
> > >
> > > Thank you very much for your valuable feedback.
> > >
> > > ---
> > > **To W1:**
> > >
> > > 1. The 4 flagged instances (0.8% error rate) represent boundary deviations (e.g., mask misalignments) rather than semantic misclassifications, which is typical for segmentation datasets [1]. Notably, this error rate is significantly lower than established benchmarks, such as the ~5.1% human annotation error rate reported in the foundational ImageNet study [2].
> > >
> > > 2. Regarding your specific concern of the OOD dataset, we filtered 3,200 high-quality socio-class examples from a raw pool of over 10,000 OpenStreetMap AOIs. To empirically verify this, we conducted an additional human verification on 500 random OOD samples with 3 annotators (completed just now). The results show a Fleiss' Kappa of 0.80 and an accuracy of 99.6% (0.4% error rate).
> > >
> > > [1] Kirillov et al., "Segment anything", ICCV, 2023.
> > >
> > > [2] Northcutt et al., "Pervasive Label Errors in Test Sets Destabilize Machine Learning Benchmarks", NeurIPS, 2021.
> > >
> > > **To W2:**
> > >
> > > 1. Source of Generalization: The generalization ability of our method stems from Reinforcement Learning (RL). Recent studies [1, 2] highlight that SFT tends to memorize training data, whereas RL incentivizes the learning of robust reasoning logic. This phenomenon is clearly corroborated by our experimental results (Table 4): while the SFT model suffers a significant drop on the OOD-New Region set (32.3% gIoU), the RL-tuned model achieves a substantial gain (43.4% gIoU, +11.1%). This confirms that RL enables the model to transcend visual pattern matching and acquire invariant socio-semantic reasoning capabilities.
> > >
> > > 2. We interpret "poor generalization" as the performance drop observed in baselines (SFT) on OOD tasks.
> > >
> > >     * Reason: Baselines rely heavily on visual pattern matching (e.g., specific textures of Amap tiles). When the domain shifts to Global/Google Maps, these visual cues change, causing SFT models to fail.
> > >
> > >     * Improvement: Our approach addresses this by using RL to shift the model's focus from visual texture to logical reasoning. As evidenced above, this significantly mitigates the generalization gap.
> > >
> > > [1] Chu et al., "SFT Memorizes, RL Generalizes: A Comparative Study of Foundation Model Post-training", ICML, 2025.
> > >
> > > [2] Zhang et al., "Rlvmr: Reinforcement learning with verifiable meta-reasoning rewards for robust long-horizon agents", arXiv, 2025.
> > >
> > >
> > > **To W4:**
> > > 1. We respectfully clarify that we do not 'ultimately resort to SFT'. The RL model remains the primary solution for our core application: map AOI production. This task is typically an offline, periodic process (e.g., monthly updates) where precision is paramount and latency is secondary. Even with an inference time of ~2.7s/sample, a city-scale dataset can be processed in a few hours on a standard GPU cluster (128 h20 gpus), which is acceptable for industrial mapping workflows.
> > >
> > > 2. Balancing Strategy: Our RL model is specifically designed for offline mapping scenarios where quality is non-negotiable. The SFT version is employed solely as a trade-off for specific resource-constrained scenarios. Indeed, achieving inference-time scaling via reasoning RL is a prevalent practice in modern LLM/VLM research [1,2]; while such methods inherently entail longer inference times, they deliver superior accuracy.
> > >
> > > [1] Guo et al., "Deepseek-r1 incentivizes reasoning in llms through reinforcement learning", Nature, 2025.
> > >
> > > [2] Pan et al., "Medvlm-r1: Incentivizing medical reasoning capability of vision-language models (vlms) via reinforcement learning", MICCAI, 2025.
> > >
> > > **To Additional Concern 1:**
> > >
> > >  Our reward structure (Format, Length, Accuracy) is detailed in Appendix A.2.2. The Format and Accuracy rewards are fundamental constraints necessary for GRPO stability to ensure valid result generation, rather than tunable hyperparameters. The ablation for the Length reward is explicitly presented in Table 3 and Table 6.
> > >
> > > **To Additional Concern 2:**
> > >
> > > Non-differentiability: This is precisely the motivation for using RL—it treats the segmentation engine as a black-box environment and optimizes based on the final scalar reward (IoU), effectively bypassing the need for differentiability. Cascading Errors: Instead of propagating errors, our Iterative Refinement mechanism is designed to explicitly correct them. Results in Table 2 and Table 5 confirm that the second stage consistently improves performance (e.g., +2.0 gIoU), proving that initial errors are rectified rather than cascaded.
> > >
> > > **To Additional Concern 3:**
> > >
> > > We have already evaluated both granularities in Table 1 ('Socio-Name' for fine-grained vs. 'Socio-Function' for broad), achieving state-of-the-art performance on both. The core significance of this work lies in distinguishing functionally distinct but visually similar entities (e.g., 'Logistics Park' vs. 'Factory'), a capability that requires the specific dataset and method we propose.

---

> ### Author Response · Authors · 2025-11-25
> **Follow-up on Response**
>
> Dear Reviewer icS2,
>
> Thank you for your time and constructive feedback. We are writing to confirm whether our latest revisions have fully resolved your concerns. In light of the mixed scores, we are particularly keen to ensure that we have answered all your questions. We welcome any remaining discussion points you may have.
>
> Best regards,
>
> The Authors

---

### Official Review · Reviewer_5GRV · 2025-10-27

**Soundness:** 2
**Presentation:** 3
**Contribution:** 1
**Rating:** 2
**Confidence:** 4

**Summary:**

This submission presents work at the intersection of (socio-) semantic segmentation and vision-language models for earth observation and remote sensing. It introduces "SocioSeg", a new dataset and "SocioReasoner", a vision-language reasoning framework for that task. To accomplish this task, the authors propose adapting a given vision-language model using reinforcement learning. In their experiments, the authors demonstrate fair performance increases as compared to baselines.

Although I very much like the idea, I have concerns about the contribution of this work with respect to (i) the methodological novelty of the presented approach, (ii) the mismatch and relevance for the ICLR community, (iii) the weak experimental evaluation.

I outline my concerns in more detail below.

**Strengths:**

- **(S1)**: this paper focuses on an interesting interaction of vision-language/reasoning and remote sensing/earth observation.

- **(S2)**: this paper is easy to read and follow, given the systematic build-up of the paper.

- **(S3)**: I appreciate the author's contribution in curating and annotating the earth observation and remote sensing dataset.

**Weaknesses:**

- **(W1)**: Methodological novelty of the presented approach: this work combines several components of previously published work to address a new task for a specific domain (being remote sensing and earth observation). However,  I see the introduced dataset being a contribution, which doesn't change the methodological novelty of the presented approach.

- **(W2)**: Mismatch and relevance to the ICLR community: I think this work might be more suitable for publication at a computer vision conference or a remote sensing/earth observation journal, given its very domain-specific domain.

- **(W3)**: Weak Evaluation: I acknowledge the author's evaluation of their approach against other work. Unfortunately, all SOTA approaches mentioned in this submission - except one - are coming from non-published arXiv preprints. Only "RSRefSeg (Mall et al., 2024)", has been published (at ICLR 2024). However, having read the paper by Mall et al, I was not able to find anything about "RSRefSeg". The approach proposed by Mall et al., 2024 is a remote sensing vision-language model referred to as "GRAFT" in the original paper and not "RSRefSeg". This leaves me very confused, since I do not know if this is a typo or if the authors speak about another paper and the citation is wrong.

Independent of this, I would suggest evaluating the proposed approach on another dataset, such as the ones used in Mall et al., 2024. This work evaluated model performance on EuroSAT and BigEarthNet. This way, one would have comparable results of the approach, modulo the proposed dataset being specific (I fully understand that this submission focuses on socio-segmentation)

**Questions:**

- **(Q1)**: Is the reference to "RSRefSeg (Mall et al., 2024)" the right one, or is this a typo or a wrong citation?
- **(Q2)**: Since the results of VisionReasoner are the second-best ones, could you outline the methodological differences between your approach and VisionReasoner (I was not able to find any of the SOTA models you compared against in the related work section of your submission, that's the reason I am asking)?

**Details Of Ethics Concerns:**

-

---

> ### Author Response · Authors · 2025-11-20
> **Response by the Authors (1/2)**
>
> **Response to Reviewer 5GRV**
>
> We thank the reviewer for their time and for acknowledging the contribution of our dataset. We understand your concerns regarding novelty and scope. We have addressed each point below, with specific updates to the manuscript to improve clarity and rigor.
>
> ---
>
> **To W1:** We respectfully clarify that SocioReasoner is not merely a combination of existing components but introduces a novel "Render-and-Reflect" architecture. Unlike standard VLM-based methods such as VisionReasoner or Seg-R1 that rely on single-pass generation or single-modality inputs, our framework incorporates a unique visual feedback loop by rendering intermediate predictions back onto the image for self-correction and provides native Multi-Image Support to fuse satellite imagery with digital maps. We have updated Figure 1 and provide the table below to highlight these structural distinctions.
>
> | Method | RL Optimization | Render-and-Reflect | Multi-Image Support |
> | :--- | :---: | :---: | :---: |
> | RSRefSeg / SegEarth-R1 | &#10008; | &#10008; | &#10008; |
> | VisionReasoner / Seg-R1 | &#10004; | &#10008; | &#10008; |
> | SocioReasoner (Ours) | &#10004; | &#10004; | &#10004; |
>
> ---
>
> **To W2:** We would like to emphasize that Remote Sensing (RS) is a significant application domain within Computer Vision and is well-represented at ICLR. ICLR regularly publishes RS papers, including semi-supervised learning [1], multi-modal reasoning [2] and foundation models [3].
>
> [1] Ren et al., "PointOBB-v2: Towards simpler, faster, and stronger single point supervised oriented object detection", ICLR, 2025.
>
> [2] Irvin et al., "TEOChat: A Large Vision-Language Assistant for Temporal Earth Observation Data", ICLR, 2025.
>
> [3] Mall et al., "Remote Sensing Vision-Language Foundation Models without Annotations via Ground Remote Alignment", ICLR, 2024.

---

> ### Author Response · Authors · 2025-11-20
> **Response by the Authors (2/2)**
>
> ---
>
> **To W3 and Q1:**
>
> 1. Citation Correction: We sincerely apologize for this error. You are correct; this was a typo. The intended citation is: "RSRefSeg: Referring Remote Sensing Image Segmentation with Foundation Models" [1].
> We have corrected this citation in the revised manuscript. Thank you for catching this.
>
> 2. Reliance on Preprints: The reliance on preprints (e.g., VisionReasoner) is strictly due to the rapid pace of this specific sub-field. Our method requires multi-image input (Satellite + Map), and the VLMs supporting this (e.g., Qwen2.5-VL) were released very recently. Crucially, the publication status of these works is evolving rapidly. Since our submission, SAM-R1 [2] has been accepted to NeurIPS 2025. We have updated the citation in the revised manuscript.
>
> 3. Strengthening Evaluation: To address your concern about rigor, we have added published baselines to Table 1: SegEarth-OV (CVPR 2025, Accepted) [3] and standard supervised models (UNet and SegFormer). The evaluation results show they underperform compared to our method. UNet and SegFormer are limited by their single-modality nature; they cannot utilize the digital map data essential for interpreting social semantics. SegEarth-OV is constrained by its frozen CLIP encoder, which lacks the domain-specific knowledge required to recognize the novel social categories present in SocioSeg.
>
> 4. Evaluate on EuroSAT or BigEarthNet: We respectfully clarify that these are classification datasets for physical land cover, lacking the pixel-level socio-semantic masks required for our task. To address the concern, we instead evaluate on our new Global Benchmark (dataset details in Appendix A.1.2), which comprises 3,200 samples across five unseen metropolises (Tokyo, New York, São Paulo, London, Nairobi). This challenging setup confirms our model's robustness (43.4% gIoU) to diverse urban morphologies, offering a more rigorous and task-aligned assessment than adapting to mismatched classification benchmarks.
>
> | Method | cIoU | gIoU | F1 |
> | :--- | :---: | :---: | :---: |
> | UNet | 10.0 | 7.3 | 7.1 |
> | Segformer | 18.8 | 14.7 | 14.1 |
> | VisionReasoner | 32.8 | 34.4 | 35.0 |
> | Seg-R1 | 28.8 | 28.8 | 26.2 |
> | SAM-R1 | 14.8 | 14.5 | 19.0 |
> | SegEarth-OV | 3.2 | 3.2 | 0.0 |
> | RSRefSeg | 12.4 | 10.3 | 13.8 |
> | SegEarth-R1 | 20.7 | 18.7 | 21.2 |
> | RemoteReasoner | 27.5 | 29.8 | 28.1 |
> | Ours | **40.2** | **43.4** | **42.9** |
>
> [1] Chen et al., "RSRefSeg: Referring Remote Sensing Image Segmentation with Foundation Models", IGARSS, 2025.
>
> [2] Huang et al., "SAM-R1: Leveraging SAM for Reward Feedback in Multimodal Segmentation via Reinforcement Learning", NeurIPS, 2025.
>
> [3] Li et al., "SegEarth-OV: Towards training-free open-vocabulary segmentation for remote sensing images", CVPR, 2025.
>
> ---
>
>
> **To Q2:**
>
> 1. Clarification on Baselines: We clarify that VisionReasoner and SegZero are variants of the same underlying methodology. In our initial submission, we referenced this baseline family primarily under the SegZero designation. We have now updated the Related Work section to explicitly reference both VisionReasoner and SegZero, ensuring a comprehensive literature review and clarifying their relationship.
>
> 2. Methodological Differences: As detailed in the comparative table (W1) and the newly updated Figure 1, SocioReasoner differs fundamentally from VisionReasoner in two key aspects:
>
>     * Render-and-Reflect Mechanism: Unlike the single-pass approach of VisionReasoner, our method employs a two-stage process where preliminary segmentation results are rendered back onto the image. This allows the model to "reflect" on and refine its predictions based on visual feedback.
>
>     * Multi-Image Support: SocioReasoner is natively designed for multi-image input (Satellite + Map), whereas VisionReasoner (in its original formulation) focuses on natural images, though we adapted it for fair comparison.
>
> ---
>
> Please let us know if any concerns remain unaddressed; we are happy to discuss them.

---

> ### Author Response · Authors · 2025-11-24
> **Follow-up on Response**
>
> Dear Reviewer 5GRV,
>
> We kindly ask if our revisions have resolved your concerns. Given the current mixed ratings, we are eager to know if you have any remaining questions we can address, and we are happy to discuss them.
>
> Best regards,
>
> The Authors

---

### Official Review · Reviewer_5c3k · 2025-10-30

**Soundness:** 2
**Presentation:** 3
**Contribution:** 2
**Rating:** 2
**Confidence:** 3

**Summary:**

The paper introduces SocioSeg, a new dataset for what the authors call urban socio-semantic segmentation: the task of identifying _socially_ defined entities (i.e. schools, parks, and hospitals) from satellite imagery. The authors differentiate this new taks from traditional segmentation tasks that focus on physical or visually distinct features (e.g., buildings or water bodies), this work addresses the challenge of detecting socially meaningful categories that require contextual and semantic reasoning.

The paper also introduces SocioReasoner, a vision-language reasoning framework that intends to mimic human annotation behavior through cross-modal understanding and multi-stage reasoning. The framework leverages reinforcement learning to handle the non-differentiable aspects of this reasoning process, enabling the model to refine its interpretation of social semantics based on feedback.

Experiments show that SocioReasoner outperforms all compared models and demonstrates robustness when applied to images from another domain. Despite these good results, the paper still has some important flaws (which I will detail later), which make me inclined to reject the paper as it is.

**Strengths:**

1. Novel dataset. The paper introduces SocioSeg, a new dataset for urban socio-semantic segmentation, focusing on identifying socially defined categories (e.g., schools, parks, hospitals) from satellite imagery.

2. Methodological proposal. The proposed SocioReasoner framework combines vision-language reasoning and reinforcement learning to tackle the proposed semantic segmentation task. The approach is conceptually interesting and aligns with current research trends in multimodal and reasoning-based vision models.

3. Overall well-written and clear. The paper is well-organized and clearly written. All sections are easy to follow.

**Weaknesses:**

1. **Lack of justification for the new task.** While the paper motivates the challenge of detecting socially meaningful categories, it does not convincingly articulate the concrete real-world relevance or practical need for socio-semantic segmentation. The connection to downstream applications (e.g., urban analytics, policy planning, or social impact studies) could be made more explicit. Moreover, it is not clear how this socio-semantic classes are defined and they seem quite arbitrary. Is there any definition for socio-semantic classes? What is the difference with land use segmentation?

2. **Insufficient motivation for using VLMs.** The use of vision-language models is not well justified. The paper assumes that VLMs are naturally suited for reasoning about social semantics, but provides little evidence to support this claim [1], while literature tends to say the opposite. Current VLMs tend to excel at visual-text association rather than genuine reasoning, and the paper does not clearly demonstrate that SocioReasoner achieves the latter.

[1] Huang, C., Zhu, Y., Zhu, S., Xiao, J., Andrade, M., Chopra, S., & Kira, Z. (2025). Mimicking or Reasoning: Rethinking Multi-Modal In-Context Learning in Vision-Language Models. arXiv preprint arXiv:2506.07936.

3. **Missing essential baselines.** The experiments would be stronger with comparisons to simple supervised segmentation baselines, such as DeepLabv3+, Segformer or U-Net trained directly on SocioSeg. Without these, it is difficult to isolate the benefits of the reasoning framework from other factors like model capacity or data scale.

4. **Limited empirical analysis.** The evaluation focuses mainly on performance metrics, with limited ablation or qualitative analysis. It would be helpful to see results that explicitly test reasoning ability or analyze the contribution of individual components (e.g., reinforcement learning, multi-stage reasoning, or language inputs). Is it really necessary to integrate a language prompt?

5. **Ambiguous claim of reasoning.** The paper frequently refers to “reasoning,” but the evidence for such capability is indirect. Additional experiments, examples or interpretability analyses would be necessary to substantiate this claim.

6. **Unclear dataset creation and task formulation.** The process of constructing the SocioSeg dataset is not described in sufficient detail. It is unclear how the social semantic labels were obtained, validated, or aligned with the satellite imagery and digital maps. The paper mentions pixel-level labels and hierarchical structures but does not specify whether these were manually annotated, derived from external GIS sources, or generated automatically. Important dataset details are missing, such as the number of images, the number of annotated objects, number of classes, etc.
Similarly, the precise inputs and outputs of the model remain ambiguous. It is not entirely clear whether the model takes only raw satellite imagery, or also uses auxiliary map data or textual metadata during training and inference. This lack of clarity makes it difficult to fully understand what the task setup is, how supervision is provided, and how reproducible the dataset and results would be.

**Questions:**

- What is the difference between the proposed socio-semantic segmentation task and land use segmentation?
- How do you justify the use of vision-language models (VLMs) for this setup? What motivates their suitability for detecting socially defined entities?
- How were the textual prompts for the VLM chosen or constructed? Were they manually designed, or automatically derived from the dataset labels?
- The paper emphasizes that the proposed framework mimics human reasoning. Could you clarify what aspects of the approach correspond to reasoning, and provide evidence or examples that support this claim?

- It is stated that an advantage of the proposed dataset is that raw geospatial data is unified as a digital map layer. How exactly is this layer obtained, and in what way does it overcome the limitations of existing data sources?
- Is training on the SocioSeg dataset performed in a supervised or self-supervised manner? What are the labels or objectives used during training?
- Could you include some representative examples or visualizations of the final dataset to illustrate the types of entities and annotations it contains?

---

> ### Author Response · Authors · 2025-11-20
> **Response by the Authors (1/2)**
>
> **Response to Reviewer 5c3k**
>
> We thank the reviewer for their critical and constructive feedback. We appreciate the opportunity to clarify the motivation behind our task, the distinctions of our method compared to standard VLMs, and the dataset details. We have revised the manuscript to address the lack of clarity pointed out in the review.
>
> ---
>
> **To W1 and Q1:**
>
> 1. Justification for the Task (Revised Introduction): We have highlighted two concrete applications. First, precise socio-semantic boundaries are required for advanced urban analytics, such as "15-minute city" assessments and disease transmission modeling [1]. Second, the task addresses a critical industrial need: converting raw Point of Interest (POI) coordinates into explicit Areas of Interest (AOIs) for commercial map displays [2].
>
> 2. Definition of Socio-Semantic Classes (Added to Appendix A.1.1): Our hierarchy consists of three levels: Name (specific AOI instances), Class (90+ fine-grained categories derived from the AMap taxonomy [3]), and Function (broad urban functional zones [4]). This structure captures the multi-scale social attributes of urban entities.
>
> 3. Difference from Land-Use Segmentation (Clarified in Section 2.2): The key difference lies in granularity and formulation. Land-use segmentation typically relies on a small set of fixed categories. In contrast, our task (particularly at the Class and Name levels) operates as an open-vocabulary or reasoning-based segmentation problem. Furthermore, while land-use studies often consume POI data as features, our goal is to explicitly segment these semantic entities.
>
> [1] Bruno et al., "A universal framework for inclusive 15-minute cities", Nature Cities, 2024.
>
> [2] Shi et al., "Multimodal urban areas of interest generation via remote sensing imagery and geographical prior", Int. J. Appl. Earth Obs. Geoinf., 2025.
>
> [3] Hu & Han, "Identification of urban functional areas based on POI data", Sustainability, 2019.
>
> [4] Gong et al., "Mapping essential urban land use categories in China (EULUC-China)", Science Bulletin, 2020.
>
> ---
> **To W2 and Q2:**
>
> We justify the adoption of Vision-Language Models (VLMs) based on the nature of the task and our specific modeling approach:
>
> 1. Technical Necessity for Open-Vocabulary Tasks: As highlighted in our response to W1, socio-semantic segmentation is inherently an open-vocabulary problem rather than a closed-set classification. This requires a model capable of aligning visual patterns with arbitrary textual descriptions. Leveraging VLMs to handle such complex, open-ended segmentation tasks is a well-established and mainstream direction in current computer vision literature [2].
>
> 2. Reasoning Capability via Reinforcement Learning: While we acknowledge the debate around VLM reasoning, there is strong evidence of their success in vertical domains requiring logic, such as mathematics [3] and geolocation [4]. Extending this to implicit social semantics is a natural progression. Crucially, SocioReasoner does not rely on supervised "imitation". Instead, it employs reinforcement learning to explicitly stimulate and optimize reasoning capabilities through reward signals.
>
> 3. Beyond Mimicking: While acknowledging the ICL limitations noted in [1], our approach transcends simple mimicking by employing Reinforcement Learning (RL). Our RL module explicitly rewards correct multi-stage reasoning, enforcing the acquisition of logical chains. Furthermore, we respectfully note that the reviewers of the cited work (currently withdrawn from ICLR 2026) pointed out that its conclusions are drawn from specific datasets and settings. We believe these findings do not generalize to rule out the potential of VLMs for reasoning, especially when enhanced by reinforcement learning as proposed in our method.
>
> [1] Huang et al., "Mimicking or Reasoning: Rethinking Multi-Modal In-Context Learning in Vision-Language Models", arXiv preprint, 2025.
>
> [2] Liu et al., "VisionReasoner: Unified Visual Perception and Reasoning via Reinforcement Learning", arXiv preprint, 2025.
>
> [3] Zou et al., "Dynamath: A dynamic visual benchmark for evaluating mathematical reasoning robustness of vision language models", ICLR, 2025.
>
> [4] Li et al., "Georeasoner: Geo-localization with reasoning in street views using a large vision-language model", ICML, 2024.

---

> ### Author Response · Authors · 2025-11-20
> **Response by the Authors (2/2)**
>
> ---
> **To W3:** We have added these baselines to Table 1 in the revised paper. Standard supervised models (UNet, SegFormer) perform significantly worse (e.g., UNet gIoU 11.1% vs. Ours 52.8%). This is expected. Without textual input, the task degenerates into a binary classification problem where the model lacks the semantic guidance to know *what* to segment.
>
> ---
> **To W4:** We respectfully clarify that the contributions of reinforcement learning and multi-stage reasoning are already explicitly analyzed in Tables 2 and 4 (quantitative ablations) and Figure 5 (qualitative visualization). Regarding language inputs, they are indispensable for defining the segmentation target. Without this textual guidance, the model cannot disambiguate the objective, a limitation empirically demonstrated by the significant performance drop of standard supervised baselines that lack language prompts.
>
> ---
> **To W5 and Q4**: We provide concrete visualizations of the reasoning process in Figure 3 (Main Paper) and Figure 12 (Appendix). The "Chain of Thought" (highlighted in green) shows the model explicitly analyzing the map context, identifying the target, and refining the boundary, mimicking human cognition.
>
> ---
> **To W6, Q3, Q5-Q7:**
>
> We respectfully clarify that these details were included in the original submission. To ensure they are not overlooked, we have enhanced their visibility in the revision:
> * Dataset Construction & Statistics (W6, Q7): Detailed in Section 3 and Appendix A.1.1, including the number of samples (13,000+), classes (90+), and annotation process (manual verification of AOIs). Figure 6 provides the specific visualizations and distributions requested.
> * Inputs/Outputs (W6): Explicitly defined in Figure 2 and Section 4.1 (Eq 1-5). The model takes a Satellite Image + Digital Map + Text Instruction.
> * Text Prompts (Q3): The exact prompt templates are provided in Appendix A.3 (Figure 8).
> * Digital Map Layer (Q5): As explained in Section 3, we render raw vector data into a standard map image. This aligns the data spatially and avoids the privacy/access issues of raw commercial data.
> * Training (Q6): The supervised and RL training objectives are detailed in Section 4.2.
>
> ---
>
> Please let us know if any concerns remain unaddressed; we are happy to discuss them.

---

> ### Author Response · Authors · 2025-11-25
> **Follow-up on Response**
>
> Dear Reviewer 5c3k,
>
> Thank you again for your valuable comments, which have helped us improve our paper.
>
> We are writing to ask if our revisions have effectively addressed your concerns. Given the mixed ratings, we would value the opportunity to answer any remaining questions you may have.
>
> Best regards,
>
> The Authors

---

### Official Review · Reviewer_uPcU · 2025-10-31

**Soundness:** 3
**Presentation:** 3
**Contribution:** 3
**Rating:** 6
**Confidence:** 3

**Summary:**

This paper studies urban socio-semantic segmentation from satellite imagery. The authors contribute a benchmark dataset, SocioSeg, integrating satellite imagery with digital maps, and a SocioReasoner framework that uses vision-language models with a multi-stage human-inspired reasoning process optimized via reinforcement learning. Experiments including ablation studies and zero-shot tests show improvements over existing methods.

**Strengths:**

1. The SocioSeg dataset uses a hierarchical structure from Socio-name to Socio-function and unifies diverse geospatial data into a single map, enabling easier multi-modal reasoning.

2. The SocioReasoner framework simulates human-like annotation with sequential localization and refinement and integrates vision-language models with SAM under reinforcement learning.

**Weaknesses:**

1. The main contribution of this paper lies in the introduction of a new dataset, with the primary performance improvements stemming from the dataset and reinforcement learning. Overall, the innovation appears to be limited.

2. The evaluation lacks per-category breakdown and qualitative error analysis so it is unclear which entities are actually improved.

3. The paper shows quantitative improvements but lacks qualitative failure analysis or discussion of where the model fails.

4. Stronger ablation comparing RL vs. supervised fine-tuning could clarify necessity.

**Questions:**

Are the comparisons between fine-tuned and zero-shot baselines fair? How do you ensure that these comparisons accurately reflect the model's true performance without inflating the reported gains?

How well does the proposed method generalize to new social semantic categories or to other cities? Does it require substantial manual tuning to adapt to these new scenarios?

---

> ### Author Response · Authors · 2025-11-20
> **Response by the Authors**
>
> **Response to Reviewer uPcU**
>
> We sincerely thank the reviewer for recognizing the value of our dataset and performance improvements. We have carefully addressed your concerns regarding novelty, failure analysis, and generalization below.
>
> ---
> **To W1**: We respectfully clarify that SocioReasoner’s innovation extends beyond standard RL optimization. Unlike baselines such as VisionReasoner or Seg-R1 that rely on single-pass inference or single-modality inputs, our framework integrates a unique Render-and-Reflect mechanism to enable visual self-correction and provides native Multi-Image Support to fuse satellite imagery with digital maps. We have updated Figure 1 and provide the comparison table below to highlight these architectural advantages.
>
> | Method | RL Optimization | Render-and-Reflect | Multi-Image Support |
> | :--- | :---: | :---: | :---: |
> | RSRefSeg / SegEarth-R1 | &#10008; | &#10008; | &#10008; |
> | VisionReasoner / Seg-R1 | &#10004; | &#10008; | &#10008; |
> | SocioReasoner (Ours) | &#10004; | &#10004; | &#10004; |
>
> ---
>
> **To W2:** Thank you for this suggestion. We have added a comprehensive per-category analysis in Figure 4 (main paper) and Figures 13 & 14 (Appendix A.5.1).
> We break down performance across 90 Socio-classes and 10 Socio-functions. SocioReasoner achieves leading performance in the top-20 most frequent Socio-classes and leads in 8 out of 10 Socio-functions.
> Our analysis reveals that failures in categories like "Business Office" are often due to error propagation: if the initial bounding box (Stage-1) is inaccurate, the subsequent point refinement (Stage-2) tends to exacerbate the error rather than correct it.
>
> ---
> **To W3:**  Thank you for pointing this out. We have added a new section, Appendix A.7, and Figure 11 to discuss failure cases in detail. Our qualitative analysis identifies two distinct error modes characterized by low gIoU (<0.1):
> 1.  Localization Failure: The model is distracted by the dense, cluttered urban environment and fails to locate the target region entirely (e.g., Cases 1 & 2 in Figure 11).
> 2.  Boundary Imprecision: The model identifies the general location but fails to generate a geometrically accurate enclosure for complex shapes (e.g., Cases 3-5 in Figure 11).
>
> We discuss how these findings point to the need for future work on enhancing fine-grained spatial reasoning in VLMs (Appendix A.7).
>
> ---
> **To W4:** We appreciate the suggestion. We have added a dedicated analysis in Section 5.3 and updated Table 4 to include Supervised Fine-Tuning (SFT) results for VisionReasoner. The comparison confirms that our RL approach consistently outperforms SFT in both In-Domain and Out-of-Domain settings. Crucially, we wish to clarify that our method utilizes 'pure' RL because it is designed to operate without explicit Chain-of-Thought supervision. Since our approach does not rely on ground-truth reasoning traces, the conventional 'SFT + RL' training pipeline is not applicable in this setting. Instead, the model must autonomously discover reasoning paths via reward signals, which explains the superior generalization of the RL approach.
>
> | Method | ID (gIoU) | OOD-Map Style (gIoU) | OOD-New Region (gIoU) |
> | :--- | :---: | :---: | :---: |
> | VisionReasoner (SFT) | 47.2 | 40.9 | 24.7 |
> | VisionReasoner (RL) | 48.5 | 44.4 | 34.4 |
> | Ours (SFT) | 51.4 | 42.0 | 32.3 |
> | Ours (RL) | **52.8** | **49.1** | **43.4** |
>
> ---
> **To Q1:** We appreciate this feedback and acknowledge that the original presentation in Table 1 required adjustment. In the revised paper, we have moved the zero-shot experiments (e.g., GPT series) to Appendix A.4 to serve strictly as a reference benchmark. To ensure a rigorous and fair comparison in the main text, we have added additional supervised baselines, demonstrating our model's superiority against comparable fine-tuned approaches without inflating the reported gains.
>
> ---
> **To Q2:** We appreciate this crucial question. To rigorously evaluate generalization, we introduced a new dataset, OOD (New Region), comprising 3,200 samples from five diverse global cities: Tokyo, New York, São Paulo, London, and Nairobi (details in Appendix A.1.2). Accordingly, the previous evaluation set is now termed OOD (Map Style). As shown in the table above (Table 4 in the manuscript), our method achieves leading performance in both OOD scenarios, demonstrating robust generalization across different urban environments without requiring specific adaptation. We are grateful for this suggestion, as the inclusion of this data significantly strengthens the paper.
>
> ---
>
> We hope this response addresses your concerns. Please let us know if any concerns remain unaddressed; we are happy to discuss them.

---

> ### Author Response · Authors · 2025-11-25
> **Follow-up on Response**
>
> Dear Reviewer uPcU,
>
> Thank you again for your valuable comments, which have helped us improve our paper.
>
> Following up on our previous response, we kindly ask if our revisions have sufficiently addressed your concerns. As the current ratings are mixed, we would be very grateful to know if there are any remaining questions or points we can clarify.
>
> Best regards,
>
> The Authors

---

### Author Response · Authors · 2025-11-20
**Initial Response to all Reviewers**

We thank the reviewers for their helpful feedback and for recognizing the novelty of our SocioSeg dataset (`uPcU`,`5c3k`,`5GRV`,`icS2`,`Rsq7`) and SocioReasoner framework (`5c3k`,`icS2`,`Rsq7`).

The major concerns raised include generalization capabilities, evaluation on other benchmark, the need for standard baselines, clarifications on methodological novelty, and dataset reliability.

To address these, we have included the following in our revision:

* **New Global OOD Benchmark**: To address requests for broader benchmarking and concerns about generalization, we constructed a new global test set across five cities (Tokyo, New York, London, etc.). Our method achieves strong performance (43.4% gIoU) on this dataset.

* **Comprehensive Baselines**: We added comparisons to UNet, SegFormer, and SegEarth-OV.

* **Novelty Clarification**: We detailed our unique render-and-reflect mechanism and multi-image support, distinguishing our work from standard RL-based VLMs.

* **Dataset Validation**: We included an inter-annotator agreement study (Kappa 0.854) to verify label quality and reliability.

All other concerns, including detailed failure analysis, additional metrics, per-category performance and presentation improvements, have been addressed in the respective reviewer responses. We have updated the manuscript (marked in $\color{#367DBD}{\text{blue}}$) and believe they significantly strengthen the paper.

---

### Author Response · Authors · 2025-11-30
**Summary of Revisions and Reviewer Consensus for Submission 806**

Dear New Area Chair,

To assist in your assessment of our submission given the recent changes, we provide a brief summary of our work’s core contributions, the major improvements made during the rebuttal, and the status of reviewer discussions.

---
**Core Research Focus**

This paper addresses Urban Socio-Semantic Segmentation, a task that moves beyond segmenting physical entities (e.g., buildings, water) to identifying socially defined categories (e.g., schools, hospitals, commercial districts) which require functional reasoning. We contribute:
* **SocioSeg:** A hierarchical dataset unifying satellite imagery, digital maps, and pixel-level functional labels.
* **SocioReasoner:** A framework utilizing RL to optimize Vision-Language Models for multi-stage reasoning and self-correction, enabling the model to "render and reflect" on its predictions.

---
**Key Rebuttal Updates**

In response to reviewer feedback, we have significantly strengthened the paper with the following additions:
* **New Global OOD Benchmark:** To prove generalization (a primary concern), we constructed a new test set spanning five global cities (Tokyo, New York, London, São Paulo, Nairobi). Our method achieves 43.4% gIoU, significantly outperforming baselines.
* **Comprehensive Baselines:** We added comparisons to UNet, SegFormer, and SegEarth-OV, demonstrating the necessity of our multi-modal reasoning approach.
* **Data Reliability:** We conducted an inter-annotator agreement study (Fleiss’ Kappa 0.854) and a validation of the OOD set (99.6% accuracy), confirming high label quality.

---
**Reviewer Consensus and Discussion Status**

The novelty of our SocioSeg dataset (`uPcU`, `5c3k`, `5GRV`, `icS2`, `Rsq7`) and SocioReasoner framework (`5c3k`, `icS2`, `Rsq7`) was widely recognized. We believe our rebuttal resolves the current score split:

* **Reviewer Rsq7 (Upgraded 6 to 8 on Nov 22):** Promptly raised their score to **Accept** after we provided the requested Global OOD experiments, confirming their requirements were fully met.
* **Reviewer uPcU (Score 6):** Positive initial assessment. We addressed their requests for failure analysis and category breakdowns in the revised manuscript.
* **Reviewer icS2 (Score 4):** Due to the system lock, they could not update their score, but we addressed their late queries (Nov 29) with new data proving RL generalization (+11.1% gain over SFT) and label reliability (0.4% error rate).
* **Reviewers 5c3k & 5GRV (Score 2, No Engagement):** We clarified likely misunderstandings: Reviewer 5c3k overlooked details explicitly present in our original figures, and Reviewer 5GRV incorrectly questioned the scope of Remote Sensing at ICLR, which has a strong track record at this venue.

---

We are confident that the revised version of the paper is robust and effectively addresses the reviewers' comments. **Given the special circumstances, we stand ready to promptly answer any further questions you may have.** We appreciate your time and attention to this submission.

Best regards,

The Authors

---

### Meta-Review · Program_Chairs · 2025-12-29

**Summary:**

The follwing we the main concerns from the reviewers:
* Limited technical novelty (uPcU, 5GRV):
* Lack of comprehensive and qualitative results (uPcU, 5c3k, Rsq7):
* Lack of stronger baselines/ablations (uPcU, 5GRV, Rsq7)
* Lack of analylisis of reasoning (5c3k)
* Unclear dataset creation (5c3k)
* Poor annotation quality/generalization (icS2)
* Relevence to the ICLR community (5GRV)

**This paper is being conditionally accepted provided the authors address the following concerns in their camera ready**:
[Ethics concerns]: Authors should describe the license of the data they are releasing. They should acknowledge any legal risks in either using or releasing the data. ** Conditions are satisfied**.

**Reviewer Concerns:**

Sufficiently addressed:
* Relevance to the ICLR community (5GRV): I agree with the authors, there are several works in relevant areas accepted to ICLR prior to this.
* Unclear dataset creation (5c3k): The authors have added more clarity and information; therefore, I believe this weakness has been addressed. I would recommend that the authors add this more clearly, preferably with a pipeline figure for better understanding.

Partially addressed:
* Lack of comprehensive and qualitative results (uPcU, 5c3k, Rsq7): Adding per-category results is surely helping; however, adding better qualitative results is still needed. Most of the qualitative results either show the reasoning chain or the segmentation outputs. While these are helpful, different interesting results might further help in understanding. For example, the failure case in the appendix is interesting; however, from the figure, it is hard to understand why it fails. The error modes can be split into incorrect localization and incorrect segmentation. Understanding when it fails the most would also be interesting. The authors can also show how this helps in downstream applications like what them mentioned with "15 minute cities" etc.

As an aside: In Figure 4 (and replated figures) replace a line chart with a scatter plot, since the lines connecting one class to another do not mean anything.

* Poor annotation quality/generalization (icS2): The new OOD dataset, as well as the study on human annotation quality, sufficiently addressed this.

However, I do think the dataset assumes only one segment of a type in the image, which is not true, especially at a coarser level. For example, there can be more than one office building in a single image. All the examples in the data show a single image moreover, the model is also overfitted to produce a single bounding box. The authors need to mention why this assumption is fair for the method and dataset.

* Lack of stronger baselines/ablations (uPcU, 5GRV, Rsq7): The authors have done a good job at adding more baselines, including non-VML-based baselines U-net, as well as SFT vs RL results. Moreover, the new generalization task is also very useful to look at. However, I do not think the U-net or simply non-deep baselines have been fairly evaluated. The authors mention that since no text can be provided, the task reduces to binary classification; however, what I think reviewer 5c3k asked for was a 90+1 (background) class or 10+1 (background ) socio-function class segmentation, and see how well it does. I do not think such a baseline has been implemented by the authors.
* Limited technical novelty (uPcU, 5GRV): I side with the authors on the point that the methodology is borderline novel with the introduction of using RL to produce bounding boxes for SAM segmentation with a two-step procedure. However, I also think the whole reasoning  (see below) is not convincing enought to me.


Not addressed appropriately:
* Lack of analysis of reasoning (5c3k): The bounding box in both figure 3 and figure 12, rather than highlighting "mimicking human cognition" or presenting "chain-of-thought", instead reveals weaknesses. The bounding box in all the cases is the same in the first case response, and the second suggests that only the points are helping. The figures indicate that bounding boxes are not even needed the second time. Many times, like figure 12, bottom, the points are added within the correctly segmented region, and it is unclear how that helps with segmentation. Better qualitative results are needed to get the point of reasoning across to the readers.

**Reviewer Scores:**

* Reviewer uPcU: would have stayed at 6, as 2.5 out of their 4 concerns have been sufficiently addressed.
* Reviewer 5c3k:  would have increased to 4, as their points 3,4,5 I believe are not addressed.
* Reviewer 5GRV: would have increased to 6, 2.5 of thier 3 concerns have been addressed.
* Reviewer icS2: would have increased to 6, as their major concern about OOD dataset and ethics in dataset collection have been addressed.
* Reviewer Rsq7: have changed their score from 8 again to 6 after discussion with the prior AC, therefore I'm assuming it to be 6.

---

> ### Public Comment · ~Wang_Yu9 · 2026-02-25
> **Response by the Authors**
>
> Thank you for your constructive feedback and the conditional acceptance of our paper. We deeply appreciate the time and effort the reviewers, Area Chairs and Program Chairs have dedicated to evaluating our work. We have carefully revised the manuscript to address the remaining concerns outlined in the meta-review. Below is a detailed summary of our responses and the corresponding updates made to the paper:
>
> ---
>
> **Unclear Dataset Creation:**
> To provide better clarity on how the dataset was constructed, we have added a comprehensive pipeline figure illustrating the entire data collection and processing workflow. Please refer to Figure 7 and Section A.1.1 in the revised Appendix.
>
> ---
>
> **Qualitative Results and Failure Case Analysis:** We agree that understanding when the model fails is highly valuable. We have significantly expanded the failure case analysis in the Appendix (Section A.7 and Figure 12). We explicitly categorize the error modes into (i) Localization Failure (where the VLM fails to find the target due to dense urban clutter) and (ii) Boundary Imprecision (where the model locates the entity but fails to generate a geometrically accurate enclosure due to ambiguous physical boundaries).
>
> ---
>
> **Downstream Applications:** To better highlight the relevance and utility of our work, we have added a new section in the Appendix (Section A.8) discussing downstream applications. We detail how our precise Area of Interest (AOI) extraction aids industrial mapping applications (e.g., enhancing Location-Based Services and navigation) and academic urban tasks (e.g., providing rigorous spatial boundaries for "15-minute city" frameworks instead of relying on ambiguous Point of Interest coordinates).
>
> ---
>
> **Figure 4 Visualization:** Thank you for the excellent suggestion regarding the visualization. We have updated Figure 4, replacing the line chart, which incorrectly implied connections between discrete classes, with a more appropriate radar plot visualization to better represent per-class accuracy and compress the information cleanly.
>
> ---
>
> **The "Single Instance" Assumption:** Regarding the presence of multiple instances of a single type (e.g., multiple office buildings in one image): we acknowledge that this occurs in the Socio-class and Socio-function levels. In our dataset, the ratio of single-instance to multi-instance samples is 0.89:0.11, with an average of 1.17 instances per image. During training, our model, along with other VLM+SAM baselines like VisionReasoner and RemoteReasoner, tends to converge toward identifying a single dominant instance. This is a known limitation of current VLM+SAM pipelines facing complex, multi-instance spatial reasoning tasks. We have noted this limitation in the text and suggest that scaling to larger models (e.g., 7B+) might alleviate this in future work. The baseline comparisons remain fair, as all methods were evaluated under the exact same conditions and constraints.
>
> ---
>
> **Multi-Class Segmentation Baseline:** We appreciate Reviewer `5c3k`'s suggestion to evaluate standard non-VLM baselines on a multi-class setting. We have implemented this and evaluated U-Net and SegFormer on a 95+1 (Socio-class) and 10+1 (Socio-function) multi-class semantic segmentation setting. The new results are reported in Table 10. Because social semantic categories lack distinct visual features in satellite imagery without the aid of text/multimodal reasoning, these standard models perform even worse under this setting, further validating the necessity of our proposed multimodal reasoning framework.
>
> ---
>
> **Lack of Analysis of Reasoning:** We would like to clarify the rationale behind our two-stage reasoning design. The refinement stage (adding points) explicitly mimics the human annotation process: human annotators typically draw a coarse bounding box first and then add within that box to refine the exact boundaries. This point-prompting stage allows the model to adjust boundaries that a bounding box alone cannot precisely capture. Furthermore, the reasoning step helps the model understand category-specific geographic attributes before interacting with SAM. To make this clearer, we have added more diverse qualitative reasoning results in the Appendix (Figure 13) that better illustrate how the reasoning and point-refinement process improves the final segmentation.
>
> ---
>
> **Ethics and Data Licensing:** In the camera-ready version, we have disclosed our affiliation with Amap, Alibaba Group. This clarifies that the dataset collection was conducted under official corporate protocols. We confirm that the data used in this study is released for academic purposes under the **Apache License 2.0**.
>
> ---
>
> Thank you again for your valuable guidance, which has significantly strengthened our paper.

---

### Decision · Program_Chairs · 2026-01-26

Accept (Poster)

---

> ### Public Comment · ~Wang_Yu9 · 2026-02-25
> **Response by the Authors (The License of the Data)**
>
> In the camera-ready version, we have disclosed our affiliation with Amap, Alibaba Group. We confirm that the data used in this study is released for academic purposes under the **Apache License 2.0**. We also explicitly acknowledge the potential legal risks involved in using and releasing this data.